
# Dissipative flow equations

**Lorenzo Rosso[1,2], Fernando Iemini[3,4], Marco Schirò[5†] and Leonardo Mazza[1⋆]**

**1** Université Paris-Saclay, CNRS, LPTMS, 91405 Orsay, France
**2** Dipartimento di Scienza Applicata e Tecnologia, Politecnico di Torino, 10129 Torino, Italy
**3** Instituto de Física, Universidade Federal Fluminense, 24210-346 Niterói, Brazil
**4** ICTP, Strada Costiera 11, 34151 Trieste, Italy
**5** JEIP, USR 3573 CNRS, Collège de France, PSL Research University,
11 Place Marcelin Berthelot, 75321 Paris Cedex 05, France
† On Leave from: Institut de Physique Théorique, Université Paris-Saclay, CNRS, CEA,
F-91191 Gif-sur-Yvette, France

⋆ leonardo.mazza@universite-paris-saclay.fr

## Abstract

We generalize the theory of flow equations to open quantum systems focusing on Lindblad master equations. We introduce and discuss three different generators of the flow that transform a linear non-Hermitian operator into a diagonal one. We first test our dissipative flow equations on a generic matrix and on a physical problem with a driven-dissipative single fermionic mode. We then move to problems with many fermionic modes and discuss the interplay between coherent (disordered) dynamics and localized losses. Our method can also be applied to non-Hermitian Hamiltonians.


# 1    Introduction

The study of many-body quantum physics has been recently challenged by the appearance of an increasing number of experimental platforms where genuine quantum phenomena take place notwithstanding the presence of an environment and of dissipation. Exciton polaritons [1, 2], lossy atomic and molecular gases [3], cavity and circuit QED arrays [4, 5], arrays of trapped ions [6] and Rydberg atoms [7], are only few prominent examples of a long list. Whereas the notion of equilibrium has been a fruitful guide to the development of standard many-body physics, these setups are inherently out of equilibrium and their description requires the introduction of a Lindblad master equation that describes in an effective way the weak coupling to a bath under the Markov approximation [8].

Several methods for addressing this dissipative out-of-equilibrium dynamics have been proposed, based for instance on quantum trajectories [9], tensor networks [10,11], extensions to mean field theories [12,13], and machine learning [14–17]; yet, the solution of many-body physics for open quantum systems remains a formidable task. Clearly, techniques developed in the framework of Hamiltonian closed systems are a continuous source of inspiration for novel developments, and in this article we present the generalization of one such technique, the so-called *flow equations* [18], to the dissipative framework.

The method of flow equations has been independently developed by Wegner in the context of condensed-matter systems [19], and by Głazek and Wilson in the context of high-energy physics [20,21]. The main idea is the search for a parameter-dependent unitary transformation that transforms the Hamiltonian into a diagonal operator where eigenvalues can be easily read out. The approach has been successfully applied to several problems; within condensed-matter physics we can briefly mention Kondo and impurity problems [18], quantum quenches in the Fermi-Hubbard model [22], quantum chemistry [23,24], quantum magnetism (including high-order perturbation theory and bound-state physics) [25–28], and more recently many-body localisation [29–38].

In this article we develop the theory of dissipative flow equations. Technically, this requires to work with generators of the real-time dynamics that are not Hermitian, whereas the established theory relied significantly on the fact that Hamiltonians are Hermitian. We propose three different kinds of flow equations that are inspired by the original contributions by F. J. Wegner [19] and S. R. White [23]. After elaborating on the links with the dissipative Schrieffer-Wolff transformation [39] (for which we present a new derivation), we show how to employ the method to infer stationary and time-dependent properties of the dynamics. We discuss several examples, three of which deal with fermionic systems, where the study of the eigenvalues of the generator of the dissipative dynamics is particularly interesting. The formulation of the flow equation for fermions requires the use of the superoperator fermionic formalism [40, 41], which is briefly reviewed in an appendix.

It is interesting to stress that attempts to using the flow equations for studying dissipative quantum systems have already appeared in the literature [42–50]. However, these approaches have typically described the global unitary dynamics of the coupled system and environment, rather than only focusing on the system, as we are proposing here. Our goal is not to follow a microscopic path, but rather to start from the beginning with a dynamics that focuses only on the system and takes into account the bath in an effective way. For this reason, our work will mainly focus on the Lindblad master equation, which is the most generic way of describing the dynamics of a system coupled to a Markovian environment. However, the method can also be used for non-Hermitian Hamiltonians.

Before concluding this introduction, it is important to stress the long-term motivation of this study. In this article we apply the dissipative flow equations only to quadratic fermionic systems, for which well-developed techniques already exist for solving the dynamics. While they perfectly serve as a benchmark for our novel method, it is also clear that our method cannot compete with them in any respect. We see this article as a first study in the direction of applying the dissipative flow equations to interacting systems which cannot be solved exactly and where approximations are necessary. Our perspective is the study of the dissipative flow equations in this context, where they could generate a new set of approximations and lead to novel solutions in a renormalization-group-like spirit (see Ref. [18] for a similar discussion in the Hamiltonian case).

The article is organized in two parts; in the former we present our theory of dissipative flow equations. In particular, in Sec. 2 we introduce the main general framework, whereas in Sec. 3 we present the details of three generators of the flow that accomplish the task of diagonalizing the Lindblad master equation in the long-flow limit. The second part is devoted to the discussion of several examples where we compare our approach with results obtained using more established techniques. In Sec. 4 we test our method on the diagonalization of a generic non-Hermitian matrix. We then move to physically-motivated problems with fermions and in Sec. 5 we discuss the problem of a single fermionic mode coupled to an environment inducing losses and gain. We then consider the problem of many fermionic modes in the presence of a localised source of losses, without disorder (Sec. 6) and with disorder (Sec. 7). This is also the occasion to discuss the dissipative flow equations in momentum space (Sec. 6) and in real space (Sec. 7). Our conclusions are presented in Sec. 8. The article is concluded by three appendices on the dissipative Schrieffer-Wolff transformation (Appendix A), on the superoperator formalism for fermions (Appendix B) and on the dissipative scattering model (Appendix C).

## 2 Dissipative Flow Equation

### 2.1 Definitions

We study the dynamics of an open quantum system in contact with a reservoir within the framework of the Markovian Lindblad master equation:

$$\frac{\mathrm{d}}{\mathrm{d}t}\rho(t) = \mathcal{L}[\rho(t)] = -\frac{i}{\hbar}[H, \rho(t)] + \sum_\alpha L_\alpha \rho(t) L_\alpha^\dagger - \frac{1}{2}\left\{L_\alpha^\dagger L_\alpha, \rho(t)\right\}. \tag{1}$$

Here, $\rho(t)$ is the density matrix of the system at time $t$, $H$ is the Hamiltonian of the system and $L_\alpha$ are the quantum-jump operators effectively describing the coupling to the environment. The superoperator $\mathcal{L}$ is linear and not Hermitian: its eigenvalues $\lambda$ are complex and such that $\Re[\lambda] \le 0$; due to its specific form, if $\lambda$ is eigenvalue then $\lambda^*$ is also eigenvalue. The eigenvalues determine the normal decaying modes of the dynamics: indeed, if $\mathcal{L}[\tau] = \lambda\tau$, then $\tau(t) = \tau \exp[\lambda t]$. The zero eigenvalue $\lambda = 0$ is particularly important because its eigenvectors represent stationary states of the dissipative dynamics: if $\mathcal{L}[\rho_0] = 0$ then $\rho(t) = \rho_0$.

Given a Lindbladian $\mathcal{L}$, we look for a parameter-dependent invertible transformation $\mathcal{S}(\ell)$ with $\ell \in \mathbb{R}^+$ such that $\mathcal{S}(0) = \mathbb{I}$ and such that

$$\mathcal{L}(\ell) = \mathcal{S}(\ell)\mathcal{L}\mathcal{S}(\ell)^{-1}, \tag{2}$$

becomes diagonal in the limit $\ell \to +\infty$. We parametrize the invertible transformation introducing a generator $\eta(\ell)$ which is a generic matrix, so that:

$$\mathcal{S}(\ell) = \mathcal{T}_\ell \exp\left[\int_0^\ell \eta(\ell')\mathrm{d}\ell'\right]. \tag{3}$$

Accordingly, it follows that

$$\frac{\mathrm{d}\mathcal{L}(\ell)}{\mathrm{d}\ell} = [\eta(\ell), \mathcal{L}(\ell)]; \qquad \frac{\mathrm{d}\mathcal{L}(\ell)^\dagger}{\mathrm{d}\ell} = -[\eta(\ell)^\dagger, \mathcal{L}(\ell)^\dagger], \tag{4}$$

where the second equality has been reported for later convenience. Clearly, $\mathcal{L}(\ell)$, $\mathcal{S}(\ell)$ and $\eta(\ell)$ are linear superoperators: they act on operators (such as the density matrix or an observable) and return an operator. In Sec. 3 we will present three generators that diagonalize $\mathcal{L}$ in the infinite-flow limit.

**Non-Hermitian Hamiltonians**

This approach can be extended to non-Hermitian Hamiltonians, which constitute the other main theoretical tool for describing open quantum systems. In this case the dynamics is described by:

$$i\hbar\frac{\mathrm{d}}{\mathrm{d}t}|\Psi(t)\rangle = \tilde{H}|\Psi(t)\rangle, \tag{5}$$

where $\tilde{H}$ is a non-Hermitian operator; its eigenvalues have a clear physical importance: real parts are related to energies and imaginary parts to gain and loss rates. The discussion proposed in the previous paragraph can be easily adapted to this situation by applying the parameter-dependent invertible transformation $S(\ell)$ to $\tilde{H}$ in order to define a non-Hermitian operator $\tilde{H}(\ell)$ that is diagonal in the infinite-flow limit (only in this specific case, $S(\ell)$ is an operator because $\tilde{H}$ is an operator). Although in the article we will explicitly consider only Lindblad master equations, all results can be easily remapped to the framework of non-Hermitian Hamiltonians; indeed, the master equations discussed in Secs. 6 and 7 require the application of the flow-equation technique to two non-Hermitian Hamiltonians.

**Invariants of the flow**

We now identify quantities that do not change during the flow $\mathcal{L}(\ell)$; the simplest example is the characteristic polynomial:

$$p_{\mathcal{L}(\ell)}(x) = \det\left[\mathcal{L}(\ell) - xI\right] = \det\left[\mathcal{S}(\ell)(\mathcal{L} - xI)\mathcal{S}(\ell)^{-1}\right] = \det\left[\mathcal{L} - xI\right] = p_{\mathcal{L}}(x); \quad (6)$$

for this reason, it is an invariant of the flow. As a consequence, every eigenvalue of the matrix $\mathcal{L}(\ell)$ is an invariant of the flow. In the following, it will be useful to use a set of quantities $I_n$ ($n \in \mathbb{N}^+$) that do not depend on $\ell$:

$$I_n = \operatorname{tr}\left[\mathcal{L}(\ell)^n\right] = \sum_i \lambda_i^n; \quad I_n \in \mathbb{C}. \quad (7)$$

It is worth mentioning that differently from the Hamiltonian setting for a generic Lindbladian $\mathcal{L}$, its 2-norm $\|\mathcal{L}(\ell)\|_2^2 = \operatorname{tr}\left[\mathcal{L}(\ell)^\dagger \mathcal{L}(\ell)\right]$ is not an invariant of the flow.

## 2.2 Observables

**Stationary-state properties**

The theory presented above gives direct access to the spectrum of $\mathcal{L}$, but in addition to that, one might be interested in computing the average of an observable $O$ over the stationary-state density matrix $\rho_0$ (we are here assuming that it is unique, but the degenerate case is treated in the same way):

$$\langle O \rangle = \operatorname{Tr}\left[O \cdot \rho_0\right]; \quad (8)$$

where we find convenient to explicitly write the symbol $\cdot$ representing the multiplication of operators. Using the fact that $\mathcal{S}^{-1}(\ell)\mathcal{S}(\ell) = 1$, we can rewrite the latter expression as

$$\langle O \rangle = \operatorname{Tr}\left[O \cdot \mathcal{S}^{-1}(\ell)\mathcal{S}(\ell)[\rho_0]\right] \equiv \operatorname{Tr}\left[O \cdot \mathcal{S}^{-1}(\ell)[\rho_0(\ell)]\right]; \quad (9)$$

where

$$\rho_0(\ell) = \mathcal{S}(\ell)[\rho_0]. \quad (10)$$

This equation, valid at any $\ell$, becomes particularly useful for $\ell \to \infty$ because the Lindbladian is diagonal and the stationary state $\rho_{0,\infty} = \lim_{\ell \to \infty} \rho_0(\ell)$ can be readily obtained:

$$\langle O \rangle = \operatorname{Tr}\left[O \cdot \mathcal{S}^{-1}(\infty)[\rho_{0,\infty}]\right]. \quad (11)$$

However, the use of Eq. (11) requires the application of $\mathcal{S}^{-1}(\infty)$ onto $\rho_{0,\infty}$, which amounts to a backward evolution of the flow and is computationally unpractical. In order to find an expression that is more computationally relevant, we observe that $\langle O \rangle$ in Eq. (11) can be expressed in terms of the Frobenius scalar product between operators: $\langle A, B \rangle_{\mathrm{F}} = \operatorname{Tr}[A^\dagger \cdot B]$; it follows that $\langle O \rangle = \langle O, \mathcal{S}^{-1}(\infty)[\rho_{0,\infty}]\rangle_{\mathrm{F}}$. By introducing the adjoint of the superoperator $\mathcal{S}^{-1}(\infty)$, we write $\langle O \rangle = \langle \mathcal{S}^{-1\dagger}(\infty)[O], \rho_{0,\infty}\rangle_{\mathrm{F}}$. The computation of the operator $O(\ell) = \mathcal{S}^{-1\dagger}(\ell)[O]$ is more practical because it obeys the differential equation:

$$\frac{\mathrm{d}O(\ell)}{\mathrm{d}\ell} = -\eta^\dagger(\ell)[O(\ell)]. \quad (12)$$

While the main flow is computed in order to put in diagonal form $\mathcal{L}$, one can use the generator $\eta(\ell)$ to evolve $O(\ell)$ with Eq. (12). Roughly speaking, this transforms the operator $O$ into the basis where the Lindbladian is diagonal.

**Time evolution**

Another interesting question concerns the real-time dynamics of the expectation value of a given operator:

$$\langle O \rangle_t = \text{Tr}[O \cdot \rho(t)], \tag{13}$$

with $\rho(t) = e^{\mathcal{L}t}[\rho(0)]$ where $e^{\mathcal{L}t}$ is the exponential of the superoperator $t\mathcal{L}$ and $\rho(0)$ is the density matrix at the initial time. We can use again the identity $\mathcal{S}(\ell)^{-1}\mathcal{S}(\ell) = 1$ and write the time evolution in the flowing basis

$$\langle O \rangle_t = \text{Tr}\big[O \cdot \mathcal{S}^{-1}(\ell) e^{\mathcal{L}(\ell)t}\mathcal{S}(\ell)[\rho(0)]\big] = \text{Tr}\big[O \cdot \mathcal{S}^{-1}(\ell)[\rho_\ell(t)]\big], \quad \rho_\ell(t) = e^{\mathcal{L}(\ell)t}\mathcal{S}(\ell)[\rho(0)]; \tag{14}$$

where we have used the following property of the operator exponential: $\mathcal{S}(\ell)e^{\mathcal{L}t}\mathcal{S}(\ell)^{-1} = e^{\mathcal{L}(\ell)t}$. Once more, the expression simplifies for $\ell \to \infty$, since in this basis the Lindbladian is diagonal. We obtain:

$$\langle O \rangle_t = \langle \mathcal{S}^{-1\dagger}(\infty)[O], \rho_\infty(t) \rangle_F. \tag{15}$$

Let us briefly analise the computational cost of this expression. While the flow is computed in order to put the Lindbladian in diagonal form, we have to compute the operator $\mathcal{S}^{-1\dagger}(\ell)[O]$ using the differential equation (12). Additionally, we must also compute the state $\rho_\ell(0) = \mathcal{S}(\ell)[\rho(0)]$, which obeys the differential equation:

$$\frac{d\rho_\ell(0)}{d\ell} = \eta(\ell)[\rho_\ell(0)]. \tag{16}$$

Whereas for computing stationary properties, we needed to only flow the observable, in this case we also need to flow the initial state. Once the flow is computed, the time-evolved state can be obtained at all times.

# 3 Generators of the flow

In this section we present three generators of the flow that accomplish the goal described in the previous section, namely the diagonalisation of the Lindbladian $\mathcal{L}$ in the infinite-flow limit $\ell \to \infty$. It will be useful to address separately the diagonal and the off-diagonal parts of the Lindbladian, that we dub respectively $\mathcal{D}(\ell)$ and $\mathcal{V}(\ell)$. Obviously, this choice is basis dependent and $\mathcal{L}(\ell) = \mathcal{D}(\ell) + \mathcal{V}(\ell)$. In the following, in order to show that the generators diagonalize the Lindbladian, we will in particular focus on $\|\mathcal{V}\|_2^2$ and on $I_2$.

## 3.1 First choice of the generator

Inspired by the original work by Wegner [18, 19], we propose the following generator of the flow:

$$\eta(\ell) = [\mathcal{L}(\ell)^\dagger, \mathcal{V}(\ell)]. \tag{17}$$

In order to show that it diagonalizes $\mathcal{L}$, we now show that it induces a flow such that $\|\mathcal{V}\|_2^2 \geq 0$ cannot increase as a function of $\ell$.

**Proof**

We first compute the derivative with respect to the flow parameter as follows:

$$\frac{d}{d\ell}\|\mathcal{V}(\ell)\|_2^2 = \frac{d}{d\ell}\text{tr}[\mathcal{V}(\ell)^\dagger \mathcal{V}(\ell)] = \text{tr}[\mathcal{V}(\ell)^\dagger \frac{d}{d\ell}\mathcal{V}(\ell)] + \text{tr}[\mathcal{V}(\ell)\frac{d}{d\ell}\mathcal{V}(\ell)^\dagger]. \tag{18}$$

Let us now discuss two interesting identities, which follow from the fact that $\mathcal{V}(\ell)$ multiplied by any diagonal operator is an off-diagonal operator:

$$\text{tr}[\mathcal{V}(\ell)\mathcal{D}(\ell)] = 0; \qquad \text{tr}\left[\mathcal{V}(\ell)\frac{\text{d}}{\text{d}\ell}\mathcal{D}(\ell)\right] = 0. \tag{19}$$

Concerning the second identity, it is important to stress that $\frac{\text{d}\mathcal{D}(\ell)}{\text{d}\ell} \neq [\eta(\ell), \mathcal{D}(\ell)]$. At this point, we can use the second of the identities (19) and write:

$$\begin{aligned}
\frac{\text{d}}{\text{d}\ell}\|\mathcal{V}(\ell)\|_2^2 &= \text{tr}[\mathcal{V}(\ell)^\dagger \frac{\text{d}}{\text{d}\ell}\mathcal{L}(\ell)] + \text{tr}[\mathcal{V}(\ell)\frac{\text{d}}{\text{d}\ell}\mathcal{L}(\ell)^\dagger] = \\
&= \text{tr}[\mathcal{V}(\ell)^\dagger[\eta(\ell), \mathcal{L}(\ell)]] - \text{tr}[\mathcal{V}(\ell)[\eta(\ell)^\dagger, \mathcal{L}(\ell)^\dagger]] = \\
&= \text{tr}[\eta(\ell)[\mathcal{L}(\ell), \mathcal{V}(\ell)^\dagger]] - \text{tr}[\eta(\ell)^\dagger[\mathcal{L}(\ell)^\dagger, \mathcal{V}(\ell)]].
\end{aligned} \tag{20}$$

If we take the definition in Eq. (17) we obtain:

$$\frac{\text{d}}{\text{d}\ell}\|\mathcal{V}(\ell)\|_2^2 = -\text{tr}[\eta(\ell)\eta(\ell)^\dagger] - \text{tr}[\eta(\ell)^\dagger\eta(\ell)] = -2\|\eta(\ell)\|_2^2 \leq 0. \tag{21}$$

**Possible issues**

Similarly to the Hamiltonian case, Eq. (21) does not rule out the possibility of a flow that does not start. This happens for instance when $\mathcal{D}(0)$ is the zero matrix. This issue can be circumvented numerically by applying at the beginning of the dynamics a random invertible operator $\mathcal{R}_0$ to the Lindbldian: $\mathcal{L}(0) \to \mathcal{R}_0\mathcal{L}(0)\mathcal{R}_0^{-1}$.

## 3.2 Second choice of the generator

As a second generator, we propose the following one, which is a slight modification of the previous one:

$$\eta(\ell) = [\mathcal{D}(\ell)^\dagger, \mathcal{V}(\ell)]; \qquad \eta_{nk}(\ell) = \left(d_{nn}^*(\ell) - d_{kk}^*(\ell)\right)\mathcal{V}_{nk}(\ell) \text{ for } n \neq k; \tag{22}$$

For later convenience, we have reported the matrix elements $\eta_{nk}$ of $\eta$ in a basis of choice, using the notation $d_{nn}$ for the diagonal matrix elements of $\mathcal{D}$ and $\mathcal{V}_{nk}$ for the off-diagonal matrix elements of $\mathcal{V}$. We do not have a proof that this generator does the desired job apart from the numerical evidence reported in the next sections, which also shows that it is more efficient than the previous generator. We can however present some considerations on its convergence based on perturbative arguments.

**Perturbative solution**

Let us begin by writing the flow equations for all matrix elements:

$$\frac{\text{d}}{\text{d}\ell}d_{nn} = \sum_q \left(\eta_{nq}\mathcal{V}_{qn} - \mathcal{V}_{nq}\eta_{qn}\right) = \sum_q 2\left(d_{nn}^* - d_{qq}^*\right)\mathcal{V}_{nq}\mathcal{V}_{qn} \tag{23a}$$

$$\begin{aligned}
\frac{\text{d}}{\text{d}\ell}\mathcal{V}_{nk} &= (d_{kk} - d_{nn})\eta_{nk} + \sum_q \left(\eta_{nq}\mathcal{V}_{qk} - \mathcal{V}_{nq}\eta_{qk}\right) = \\
&= -|d_{nn} - d_{kk}|^2 \mathcal{V}_{nk} + \sum_q \left(d_{nn}^* - 2d_{qq}^* + d_{kk}^*\right)\mathcal{V}_{nq}\mathcal{V}_{qk}.
\end{aligned} \tag{23b}$$

The solution of this set of equation looks rather demanding, but it can be simplified if we assume that at the initial time the following condition holds:

$$|\mathcal{V}_{nk}| \ll |d_{nn} - d_{kk}|. \tag{24}$$

We introduce the small parameter $\xi$ proportional to the off-diagonal part of the Lindbladian. We now expand each matrix element as:

$$d_{nn}(\ell) = d_{nn}^{(0)}(\ell) + \xi d_{nn}^{(1)}(\ell) + \xi^2 d_{nn}^{(2)}(\ell) + \dots \tag{25a}$$

$$\mathcal{V}_{nk}(\ell) = \xi \mathcal{V}_{nk}^{(1)}(\ell) + \xi^2 \mathcal{V}_{nk}^{(2)}(\ell) + \dots \tag{25b}$$

with initial conditions:

$$d_{nn}^{(0)}(0) = d_{nn}(0), \quad d_{nn}^{(m)}(0) = 0 \text{ for } m > 0; \tag{26a}$$

$$\mathcal{V}_{nk}^{(1)}(0) = \frac{\mathcal{V}_{nk}(0)}{\xi}, \quad \mathcal{V}_{nk}^{(m)}(0) = 0 \text{ for } m > 1. \tag{26b}$$

The flow equations for each term of the expansion are obtained by comparing terms of the same power of $\xi$:

$$\frac{d}{d\ell} d_{nn}^{(0)} = 0; \qquad \frac{d}{d\ell} d_{nn}^{(1)} = 0; \qquad \frac{d}{d\ell} d_{nn}^{(2)} = \sum_q \left( d_{nn}^{(0)*} - d_{qq}^{(0)*} \right) \mathcal{V}_{nq}^{(1)} \mathcal{V}_{qn}^{(1)}; \tag{27a}$$

$$\frac{d}{d\ell} \mathcal{V}_{nk}^{(1)} = -|d_{nn}^{(0)} - d_{kk}^{(0)}|^2 \mathcal{V}_{nk}^{(1)}. \tag{27b}$$

At the lowest non-trivial order for each term, the equations are solved by:

$$d_{nn}(\ell) = d_{nn}(0) + \frac{1}{2} \sum_q \frac{\mathcal{V}_{nq}(0)\mathcal{V}_{qn}(0)}{d_{nn}(0) - d_{qq}(0)} \left( 1 - e^{-2|d_{nn}(0)-d_{qq}(0)|^2\ell} \right) \tag{28a}$$

$$\mathcal{V}_{nk} = \mathcal{V}_{nk}(0) e^{-|d_{nn}(0)-d_{kk}(0)|^2\ell} + \dots \tag{28b}$$

We obtain a correction that is thus completely consistent with perturbation theory (although we remark that flow equations are a technique that is not perturbative in spirit). It is interesting to observe that in this limit the off-diagonal matrix elements $\mathcal{V}_{nk}$ become zero with a typical flow scale that is proportional to their eigenvalue difference $|d_{nn}(0) - d_{kk}(0)|^2 \sim |d_{nn}(\ell) - d_{kk}(\ell)|^2$. This means that an interpretation of the flow as a generalized renormalization group where the decimation suppresses terms that are distant in energies might be possible as in the Hamiltonian case [18].

## 3.3 Third choice of the generator

Let us now consider the following generator:

$$\eta_{nk}(\ell) = \begin{cases} \dfrac{\mathcal{V}_{nk}(\ell)}{\mathcal{D}_{nn}(\ell) - \mathcal{D}_{kk}(\ell)}, & \text{if } \mathcal{D}_{nn}(\ell) \neq \mathcal{D}_{kk}(\ell); \\ 0, & \text{if } \mathcal{D}_{nn}(\ell) = \mathcal{D}_{kk}(\ell). \end{cases} \tag{29}$$

which resembles the one proposed by S. White in the Hamiltonian setting [23].

**Perturbative limit and dissipative Schrieffer-Wolff transformation**

Even if we are going to present a proof that shows that this generator works also in non-perturbative regimes, we motivate its introduction considering the case of an off-diagonal part of $\mathcal{L}(\ell)$ that is a small perturbation; its strength is controled by a small parameter $\xi$. This situation is not uncommon in open quantum systems. For instance, in Ref. [51] the authors discuss a perturbative approach to a ring of spins subject to strong dissipation and to a perturbatively small spin-spin coherent coupling. The same authors discuss in Ref. [52] an open Jaynes-Cumming lattice where the spin-photon coupling is small and treated perturbatively.

We thus propose an expansion of the generator $\eta(\ell)$ in powers of $\xi$:

$$\eta(\ell) = \xi\eta^{(1)}(\ell) + \xi^2\eta^{(2)}(\ell) + \dots \tag{30}$$

No zero-th order term is introduced because for $\xi = 0$ the matrix is diagonal and the generator $\eta(\ell)$ must be equal to zero, so that $\mathcal{S}(\ell) = I$. At first order in $\xi$, the flow equation reads:

$$\frac{d}{d\ell}\mathcal{L}(\ell) = [\xi\eta^{(1)}(\ell) + \xi^2\eta^{(2)}(\ell) + \dots, \mathcal{D}(\ell) + \mathcal{V}(\ell)] = \xi[\eta^{(1)}(\ell), \mathcal{D}(\ell)] + o(\xi^2). \tag{31}$$

In order to kill the off-diagonal part of $\mathcal{L}(\ell)$, it is reasonable to ask:

$$\xi[\eta^{(1)}(\ell), \mathcal{D}(\ell)] = -\mathcal{V}(\ell). \tag{32}$$

This latter equation defines the third generator of the dynamics, whose matrix elements were presented in Eq. (29).

It is important to observe that Eq. (32) also allows us to establish a link with the theory of dissipative Schrieffer-Wolff transformation, whose form was first derived in Ref. [39], and for which we present another derivation in Appendix A. In this approach, one looks for an invertible transformation that rotates the Hilbert space and decouples subspaces related to different eigenvalues of the unperturbed system. It is customary to introduce a generator also for the Schrieffer-Wolff transformation, and at first order in the strength of the perturbation one obtains exactly Eq. (32) (see Eq. (102) in Appendix A). Thus, the third generator implements a flow that reproduces the action of the Schrieffer-Wolff transformation.

**Proof**

We consider the second invariant of the flow:

$$I_2 = I_2^{\text{diag}}(\ell) + I_2^{\text{off}}(\ell) \quad \text{with} \quad I_2^{\text{diag}}(\ell) = \sum_n \mathcal{L}_{nn}^2(\ell); \quad I_2^{\text{off}}(\ell) = \sum_{n \neq m} \mathcal{L}_{nm}(\ell)\mathcal{L}_{mn}(\ell). \tag{33}$$

We now study the derivative with respect to $\ell$ of $I_2^{\text{off}}$: using the fact that $\frac{d}{d\ell}I_2 = 0$ we obtain

$$\frac{d}{d\ell}I_2^{\text{off}} = -2\sum_n \mathcal{L}_{nn}(\ell)\frac{d}{d\ell}\mathcal{L}_{nn}(\ell) = -2\sum_{n \neq k}\mathcal{L}_{nn}(\ell)\big(\eta_{nk}(\ell)\mathcal{L}_{kn}(\ell) - \mathcal{L}_{nk}(\ell)\eta_{kn}(\ell)\big) =$$

$$= -2\sum_{n \neq k}\mathcal{L}_{nn}(\ell)\frac{\mathcal{L}_{nk}(\ell)\mathcal{L}_{kn}(\ell) + \mathcal{L}_{nk}(\ell)\mathcal{L}_{kn}(\ell)}{\mathcal{L}_{nn}(\ell) - \mathcal{L}_{kk}(\ell)} =$$

$$= -\sum_{n \neq k}\big(\mathcal{L}_{nn}(\ell) - \mathcal{L}_{kk}(\ell)\big)\frac{\mathcal{L}_{nk}(\ell)\mathcal{L}_{kn}(\ell) + \mathcal{L}_{nk}(\ell)\mathcal{L}_{kn}(\ell)}{\mathcal{L}_{nn}(\ell) - \mathcal{L}_{kk}(\ell)} = -2I_2^{\text{off}}. \tag{34}$$

This differential equation is solved by:

$$I_2^{\text{off}}(\ell) = I_2^{\text{off}}(0)e^{-2\ell}, \tag{35}$$

and shows a very efficient approach to 0 with a typical flow-scale that does not depend on any system parameter. This scaling is expected to be particularly useful in numerical applications.

**Possible issues**

The fact that $I_2^{\text{off}}$ is equal to zero does not mathematically guarantee that the off-diagonal part of the Lindbladian matrix is equal to zero; one possible example is constituted by an initial matrix that is triangular: in this case $I_2^{\text{off}}(0) = 0$. This issue, that is not present in the Hamiltonian case, can be circumvented numerically by applying at the beginning of the dynamics a random invertible operator $\mathcal{R}_0$ to the Lindbladian: $\mathcal{L}(0) \to \mathcal{R}_0\mathcal{L}(0)\mathcal{R}_0^{-1}$.

# 4  First example: Numerical tests on a generic matrix

As a first example, we apply the flow equations to a generic non-Hermitian complex matrix $A$ with size $15 \times 15$; the real and imaginary parts of the matrix elements are randomly taken from the uniform distribution in the range $[-1, 1]$. We have verified that the main qualitative features of the data that we present in this Section do not depend on the specific choice of $A$.

   We implement the numerical solution by means of a 5-th order Runge-Kutta algorithm (Butcher's scheme); the flow step is $d\ell = 10^{-3}$ and $N_{step} = 15000$, so that $\ell_{\max} = 15$. During the flow, we set to zero the off-diagonal matrix elements whenever their absolute value is less than the 1% of the initial value, as routinely done in Hamiltonian setting [18]. We verify that at the end of the simulation the matrix $A(\ell_{\max})$ is well approximated by a diagonal matrix; the real and imaginary parts of the diagonal elements are compared with the values obtained by standard matrix diagonalization and we benchmark the convergence using the discrepancy $\Delta^2 = \sum_{i=1}^{15} |\lambda_i^{(\text{flow})} - \lambda_i^{(\text{exact})}|^2$. We also study the invariants $I_n$, which should be conserved by the flow, and in particular focus on the relative error $\delta I_n = |I_n(\ell_{\max}) - I_n(\ell = 0)|/|I_n(\ell = 0)|$.

   We study the dissipative flow equations using the first, the second and the third generator. Concretely, we promote the matrix $A$ to a parameter-dependent matrix $A(\ell)$ and solve the differential equation (see Eq. (4)):

$$\frac{dA(\ell)}{d\ell} = [\eta(\ell), A(\ell)]. \tag{36}$$

The generator $\eta(\ell)$ is constructed by splitting the matrix into a diagonal and off-diagonal part $A(\ell) = \mathcal{D}(\ell) + \mathcal{V}(\ell)$ and subsequently following the prescrptions in Eqs. (17), (22) and (29), respectively. The real and imaginary parts of the diagonal matrix elements are plotted in Figs. 1, 2 and 3, respectively; a comparison with the exact results obtained with standard diagonalization routines is also presented.

   An important property of the third generator is the fact that the quantity $I_2^{\text{off}}(\ell)$ satisfies the differential equation (35); as a consequence, the convergence to the diagonal form is very efficient. The exponential behaviour $I_2^{\text{off}}(\ell) = I_2^{\text{off}}(0)e^{-2\ell}$ has been numerically verified, see Fig. 4. A similar numerical calculation with the first and second generator shows a significantly slower approach to zero, although in both cases we eventually recover an exponential law (see Fig. 4). We thus conclude that for numerical applications the third generator is the most efficient choice.

# 5  Second example: Fermionic mode with losses and gain

In this and following Sections we test the dissipative flow equations with several physical examples for which exact solutions can be easily obtained analytically and numerically. We will show that the theory is correct and discuss the specific properties of each generator, highlighting advantages and disadvantages. We begin in this Section with the study of a single fermionic mode coupled to a bath inducing incoherent losses and gain. The interest of this simple example relies on the fact that the flow equations can be solved analytically in several limits.

## 5.1  The problem

We study a single fermionic mode with energy $\varepsilon$ coupled to a bath; fermions are lost at a rate $\Gamma_1$ and gained at a rate $\Gamma_2$. We introduce the canonical fermionic operators $\hat{c}$ and $\hat{c}^\dagger$, and model

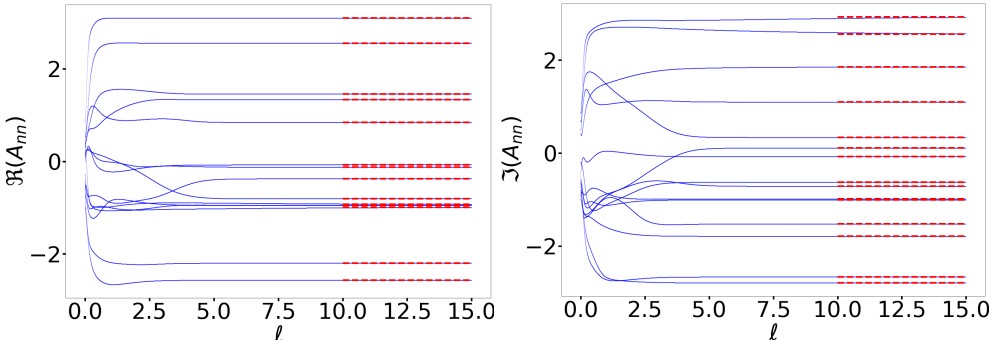

Figure 1: Flow equation solutions for the real (left panel) and imaginary (right panel) parts of the diagonal elements of the generic matrix $A$ using the first generator in Eq. (17); dashed red lines represent the eigenvalues obtained by standard diagonalization routines. For this specific case, the discrepancy is $\Delta \simeq 8.5 \cdot 10^{-3}$; we have verified that the relative error of the invariants $\delta I_n$ with $1 \leq n \leq 15$ never exceeds $10^{-5}$, the worst case being $I_{15}$.

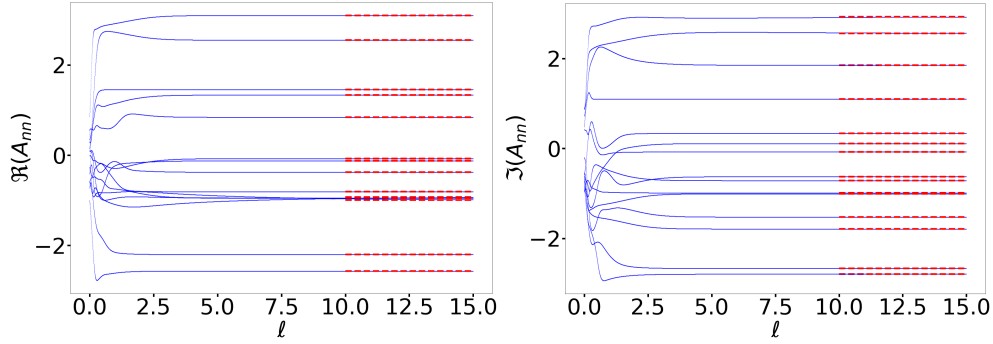

Figure 2: Flow equation solutions for the real (left panel) and imaginary (right panel) parts of the diagonal elements of the generic matrix $A$ using the second generator in Eq. (22); dashed red lines represent the eigenvalues obtained by standard diagonalization routines. For this specific case, the discrepancy is $\Delta \simeq 7.3 \cdot 10^{-3}$; we have verified that the relative error of the invariants $\delta I_n$ with $1 \leq n \leq 15$ never exceeds $10^{-8}$, the worst case being $I_{15}$.

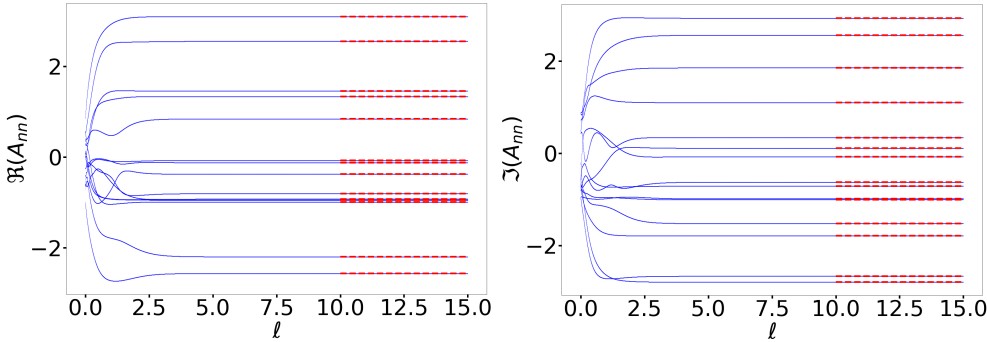

Figure 3: Flow equation solutions for the real (left panel) and imaginary (right panel) parts of the diagonal elements of the generic matrix $A$ using the third generator in Eq. (29); dashed red lines represent the eigenvalues obtained by standard diagonalization routines. For this specific case, the discrepancy is $\Delta \simeq 1.9 \cdot 10^{-7}$; we have verified that the relative error of the invariants $\delta I_n$ with $1 \leq n \leq 15$ never exceeds $10^{-7}$, the worst case being $I_{15}$.

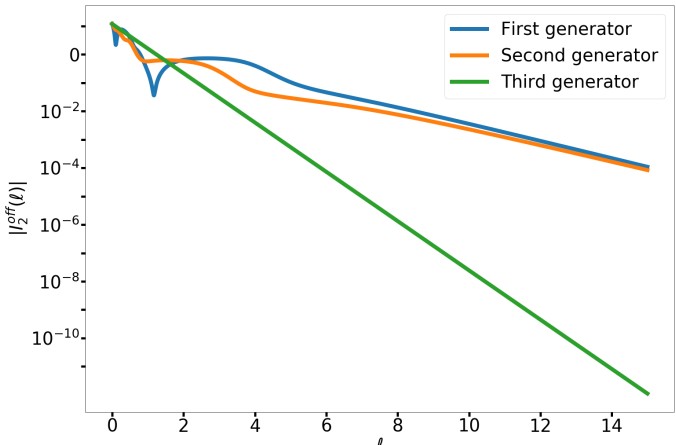

Figure 4: Behaviour during the flow of $|I_2^{\mathrm{off}}(\ell)|$ for the first (blue), second (orange) and third (green) generator. In this latter case, we recover the behaviour predicted in formula (35).

the dynamics of the site with the following master equation:

$$\frac{\mathrm{d}}{\mathrm{d}t}\rho(t) = -\frac{i}{\hbar}[H,\rho(t)] + \sum_j L_j \rho(t) L_j^\dagger - \frac{1}{2}\left\{L_j^\dagger L_j, \rho(t)\right\}; \tag{37a}$$

$$H = \varepsilon \hat{c}^\dagger \hat{c}, \quad L_1 = \sqrt{\Gamma_1}\hat{c}, \quad L_2 = \sqrt{\Gamma_2}\hat{c}^\dagger. \tag{37b}$$

We propose the study of this model using the formalism of *superfermion representation*, that is presented in detail in Refs. [40, 41] and that is also reviewed in Appendix B. Roughly speaking, the idea is to treat the density matrix as a vector of an appropriate Hilbert space isomorphic to $\mathcal{H} \otimes \mathcal{H}$, where $\mathcal{H}$ is the two-dimensional Hilbert space of a single fermionic mode: $\rho(t) \to |\rho(t)\rangle$. We subsequently need to introduce superoperators $c$ and $\tilde{c}$ that describe the action of fermionic operators on the left or on the right of the density matrix; they square to zero ($c^2 = 0$ and $\tilde{c}^{\dagger 2} = 0$) and satisfy mutual canonical anticommutation relations $\{c^{(\dagger)}, \tilde{c}^{(\dagger)}\} = 0$. To all effects, this formalism describes the dynamics of a single fermionic mode coupled to a bath as a fermionic two-mode problem. This approach has the great advantage to allow to represent a quadratic fermionic master equation as the one in Eq. (37) as a matrix, and to link the normal decaying modes to its eigenvalues.

A detailed discussion of model (37) using superoperators is reported in Ref. [40]; here we briefly mention some relevant aspects. According to this formalism, (37) can be cast in the following form:

$$i\hbar \frac{\mathrm{d}}{\mathrm{d}t}|\rho(t)\rangle = L|\rho(t)\rangle, \tag{38}$$

where $L$ is an operator that is quadratic in the fermionic superoperators:

$$L = \varepsilon\left(c^\dagger c - \tilde{c}^\dagger \tilde{c}\right) - i\hbar \frac{\Gamma_1 - \Gamma_2}{2}\left(c^\dagger c + \tilde{c}^\dagger \tilde{c}\right) + \hbar\left(\Gamma_1 c\tilde{c} + \Gamma_2 c^\dagger \tilde{c}^\dagger\right) - i\hbar\Gamma_2. \tag{39}$$

Note that roughly this expression can be obtained from (37) by using a $\tilde{c}$ operator each time the fermions act to the right of the density matrix. Since it is a quadratic operator, we can put it into a $2 \times 2$ matrix form by exploiting the aforementioned anticommutation relations. We obtain:

$$L = \begin{pmatrix} c^\dagger & \tilde{c} \end{pmatrix} \begin{pmatrix} \varepsilon - \frac{i\hbar}{2}\Delta\Gamma_{12} & \hbar\Gamma_2 \\ -\hbar\Gamma_1 & \varepsilon + \frac{i\hbar}{2}\Delta\Gamma_{12} \end{pmatrix} \begin{pmatrix} c \\ \tilde{c}^\dagger \end{pmatrix} - \varepsilon - \frac{i\hbar}{2}(\Gamma_1 + \Gamma_2), \qquad \Delta\Gamma_{12} = \Gamma_1 - \Gamma_2. \tag{40}$$

It is possible to diagonalize the matrix with an invertible and non-unitary transformation (not specified for the moment) and introducing the operators $d$, $D^\dagger$, $\tilde{D}$ and $\tilde{d}^\dagger$, so that:

$$
\begin{aligned}
L &= \begin{pmatrix} D^\dagger & \tilde{D} \end{pmatrix} \begin{pmatrix} \varepsilon - \frac{i\hbar}{2}(\Gamma_1 + \Gamma_2) & 0 \\ 0 & \varepsilon + \frac{i\hbar}{2}(\Gamma_1 + \Gamma_2) \end{pmatrix} \begin{pmatrix} d \\ \tilde{d}^\dagger \end{pmatrix} - \varepsilon - \frac{i\hbar}{2}(\Gamma_1 + \Gamma_2) = \\
&= \left( \varepsilon - \frac{i\hbar}{2}(\Gamma_1 + \Gamma_2) \right) D^\dagger d + \left( -\varepsilon - \frac{i\hbar}{2}(\Gamma_1 + \Gamma_2) \right) \tilde{d}^\dagger \tilde{D} .
\end{aligned}
\tag{41}
$$

The steady state is annihilated by both $d$ and $\tilde{D}$ operators:

$$
d|\rho_\infty\rangle = 0, \quad \tilde{D}|\rho_\infty\rangle = 0 ;
\tag{42}
$$

and is defined by this relation. The normal decaying modes are obtained by applying the $D^\dagger$ and $\tilde{d}^\dagger$ operators onto the steady state; the corresponding Lindbladian eigenvalue determines their time evolution. In this case, for instance, by looking at the eigenvalues of $L$, which are $\lambda_\pm = \varepsilon \pm \frac{i\hbar}{2}(\Gamma_1 + \Gamma_2)$, we can infer that decays take place according to the typical time $\tau = (\Gamma_1 + \Gamma_2)^{-1}$.

It is to be stressed that the eigenvalues $\lambda_\pm$ with positive or negative imaginary part are an artifact of this formalism, and do not correspond to losses and gain, but only to losses, as it should be for this problem. The reason is apparent by looking at the passage from Eq. (39) to Eq. (40), where we need to anticommute $\tilde{c}$ and $\tilde{c}^\dagger$. However, the physical significance is fully restored when putting the master equation in canonical form, as in Eq. (41).

## 5.2 Three approaches with dissipative flow equations

We now apply the techniques of the flow equations to the matrix representation of the superoperator $L$ in Eq. (40) in order to put it in diagonal form and extract the two eigenvalues $\lambda_\pm$.

We parametrize the Lindbladian matrix $L(\ell)$ in the following way:

$$
L(\ell) = \begin{pmatrix} \epsilon(\ell) + i\alpha(\ell) & \mu_2(\ell) \\ -\mu_1(\ell) & \epsilon(\ell) - i\alpha(\ell) \end{pmatrix}, \qquad \begin{cases} \alpha(\ell), \epsilon(\ell), \mu_{1,2}(\ell) \in \mathbb{R} \\ \alpha(0) = -\frac{\hbar}{2}\Delta\Gamma_{12}, \; \epsilon(0) = -\varepsilon, \; \mu_{1,2}(0) = \hbar\Gamma_{1,2}. \end{cases}
\tag{43}
$$

**Invariants of the flow**

Since the matrix is $2 \times 2$, there are only two independent invariants of the flow:

$$
\mathrm{tr}[L(\ell)] = 2\epsilon(\ell), \qquad \mathrm{tr}\left[L(\ell)^2\right] = (\epsilon(\ell) + i\alpha(\ell))^2 + (\epsilon(\ell) - i\alpha(\ell))^2 - 2\mu_1(\ell)\mu_2(\ell) .
\tag{44}
$$

From the invariance of these expressions, we obtain that:

$$
\epsilon(\ell) = \epsilon(0) = \varepsilon, \qquad \alpha^2(\ell) + \mu_1(\ell)\mu_2(\ell) = \hbar^2 \frac{(\Gamma_1 - \Gamma_2)^2}{4} + \hbar^2 \Gamma_1 \Gamma_2 = \hbar^2 \frac{(\Gamma_1 + \Gamma_2)^2}{4} .
\tag{45}
$$

**First generator**

The first generator of the dynamics reads:

$$
\eta(\ell) = [\mathcal{L}(\ell)^\dagger, \mathcal{V}(\ell)] = \begin{pmatrix} \mu_1^2(\ell) - \mu_2^2(\ell) & -2i\,\alpha(\ell)\mu_2(\ell) \\ -2i\,\alpha(\ell)\mu_1(\ell) & -\mu_1^2(\ell) + \mu_2^2(\ell) \end{pmatrix},
\tag{46}
$$

and its commutator with the Lindbladian matrix:

$$
[\eta(\ell), \mathcal{L}(\ell)] = \begin{pmatrix} 4i\alpha(\ell)\mu_1(\ell)\mu_2(\ell) & -2\mu_2(\ell)\left(2\alpha^2(\ell) - \mu_1^2(\ell) + \mu_2^2(\ell)\right) \\ 2\mu_1(\ell)\left(2\alpha^2(\ell) + \mu_1^2(\ell) - \mu_2^2(\ell)\right) & -4i\alpha(\ell)\mu_1(\ell)\mu_2(\ell) \end{pmatrix} .
\tag{47}
$$

This leads to a set of coupled non-linear differential equations:

$$\frac{\mathrm{d}}{\mathrm{d}\ell}\alpha(\ell) = 4\mu_1(\ell)\mu_2(\ell)\alpha(\ell); \tag{48a}$$

$$\frac{\mathrm{d}}{\mathrm{d}\ell}\mu_1(\ell) = -2\mu_1(\ell)\big(2\alpha^2(\ell)+\mu_1^2(\ell)-\mu_2^2(\ell)\big); \tag{48b}$$

$$\frac{\mathrm{d}}{\mathrm{d}\ell}\mu_2(\ell) = -2\mu_2(\ell)\big(2\alpha^2(\ell)-\mu_1^2(\ell)+\mu_2^2(\ell)\big). \tag{48c}$$

By using the invariants listed above, we can reduce the three equations to a single one:

$$\frac{\mathrm{d}}{\mathrm{d}\ell}\alpha(\ell) = \big[\hbar^2\,(\Gamma_1+\Gamma_2)^2 - 4\alpha^2(\ell)\big]\alpha(\ell). \tag{49}$$

We do not give an explicit (and useless) analytical solution here; it is however clear that the stationary values of $\alpha(\ell)$ are $\bar\alpha_{1,2,3} = \{\pm(\Gamma_1+\Gamma_2)/2, 0\}$, among which we find the correct value. The specific value is determined by the initial conditions. Using the second invariant, we can conclude that the stationary values $\bar\alpha_{1,2} = \pm(\Gamma_2+\Gamma_2)/2$ are accompanied by the stationary value $\bar\mu_1\bar\mu_2 = 0$; Eqs. (48) additionally say that each $\bar\mu_1$ and $\bar\mu_2$ should be equal to zero. Vice-versa, the stationary value $\bar\alpha_3 = 0$ is accompanied by $\bar\mu_1\bar\mu_2 = (\Gamma_1+\Gamma_2)^2/4$.

In order to test the stability of the three stationary solutions of the flow, we consider Eq. (49), which we write in the form $\frac{\mathrm{d}}{\mathrm{d}\ell}\alpha(\ell) = f(\alpha)$. We then evaluate $f'(\alpha_{1,2,3})$ for the three stationary values and obtain:

$$f'(\bar\alpha_{1,2}) = -2(\Gamma_1+\Gamma_2)^2 \leq 0 \qquad \implies \qquad \text{stable}; \tag{50a}$$

$$f'(\bar\alpha_3) = (\Gamma_1+\Gamma_2)^2 - 12\alpha^2 = (\Gamma_1+\Gamma_2)^2 \geq 0 \qquad \implies \qquad \text{unstable}. \tag{50b}$$

Given the presence of the second invariant, that links the flow of $\alpha(\ell)$ to that of $\mu_1(\ell)\mu_2(\ell)$, we can conclude that if $\bar\alpha_{1,2}$ is a stable stationary values, then also $\bar\mu_1 = 0$ and $\bar\mu_2 = 0$ are stable stationary values. This simple analysis reveals that the stationary points with vanishining off-diagonal part are locally stable, and depending on the initial conditions, one of the two specific value $\bar\alpha_{1,2}$ is selected.

**Second generator**

The second generator of the dynamics reads:

$$\eta(\ell) = [\mathcal{D}(\ell)^\dagger, \mathcal{V}(\ell)] = \begin{pmatrix} 0 & -2i\,\alpha(\ell)\mu_2(\ell) \\ -2i\,\alpha(\ell)\mu_1(\ell) & 0 \end{pmatrix}, \tag{51}$$

and its commutator with the Lindbladian matrix:

$$[\eta(\ell),\mathcal{L}(\ell)] = 4\alpha(\ell)\begin{pmatrix} i\mu_1(\ell)\mu_2(\ell) & -\alpha(\ell)\mu_2(\ell) \\ \alpha(\ell)\mu_1(\ell) & -i\mu_1(\ell)\mu_2(\ell) \end{pmatrix}. \tag{52}$$

This leads to the following set of coupled non-linear differential equations:

$$\frac{\mathrm{d}}{\mathrm{d}\ell}\alpha(\ell) = 4\mu_1(\ell)\mu_2(\ell)\alpha(\ell); \qquad \frac{\mathrm{d}}{\mathrm{d}\ell}\mu_i(\ell) = -4\alpha^2(\ell)\mu_i(\ell). \tag{53}$$

These equations are very similar to those obtained with the first generator and reported in Eq. (48). The stationary values can be obtained with the invariants of the flow. This approach leads us again to Eq. (49), for which the analysis discussed above can be repeated.

**Third generator**

The third generator of the dynamics reads:

$$\eta(\ell) = \begin{pmatrix} 0 & \frac{\mathcal{V}_{12}(\ell)}{\mathcal{D}_{11}(\ell)-\mathcal{D}_{22}(\ell)} \\ \frac{\mathcal{V}_{21}(\ell)}{\mathcal{D}_{22}(\ell)-\mathcal{D}_{11}(\ell)} & 0 \end{pmatrix} = -\frac{i}{2\alpha(\ell)} \begin{pmatrix} 0 & \mu_2(\ell) \\ \mu_1(\ell) & 0 \end{pmatrix}, \tag{54}$$

and its commutator with the Lindbladian matrix:

$$[\eta(\ell), \mathcal{L}(\ell)] = \begin{pmatrix} i\frac{\mu_1(\ell)\mu_2(\ell)}{\alpha(\ell)} & -\mu_2(\ell) \\ \mu_1(\ell) & -i\frac{\mu_1(\ell)\mu_2(\ell)}{\alpha(\ell)} \end{pmatrix}. \tag{55}$$

This leads to the following set of coupled non-linear differential equations:

$$\frac{\mathrm{d}}{\mathrm{d}\ell}\alpha(\ell) = \frac{\mu_1(\ell)\mu_2(\ell)}{\alpha(\ell)}; \qquad \frac{\mathrm{d}}{\mathrm{d}\ell}\mu_i(\ell) = -\mu_i(\ell). \tag{56}$$

It is very easy to obtain $\mu_i(\ell) = \mu_i(0)e^{-\ell}$ and accordingly to verify property (35), concerning the flow-evolution of $I_2^{\mathrm{off}}$:

$$I_2^{\mathrm{off}}(\ell) = -\mu_1(\ell)\mu_2(\ell) = -\Gamma_1\Gamma_2 e^{-2\ell} = I_2^{\mathrm{off}}e^{-2\ell}. \tag{57}$$

Using the second invariant, we can also obtain $\alpha(\ell)$:

$$\alpha(\ell) = -\sqrt{\frac{(\Gamma_1+\Gamma_2)^2}{4} - \Gamma_1\Gamma_2 e^{-2\ell}} \xrightarrow{\ell\to\infty} -\frac{\Gamma_1+\Gamma_2}{2}. \tag{58}$$

**A simple analytical case:** $\Gamma_2 = 0$

In order to shed more light on the technique, we consider here the special case $\Gamma_2 = 0$. In order to study this special case, we re-parametrize $L(\ell)$ setting $\mu_2(\ell) = 0$; the matrix is now triangular, its eigenvalues can be directly read from the diagonal, which is not supposed to evolve during the flow. Indeed, from the second invariant we obtain $\alpha(\ell) = -\frac{\Gamma_1}{2}$. We notably simplifies the differential equations satisfied by $\mu_1(\ell)$; we list them here below for the three generators:

$$\frac{\mathrm{d}}{\mathrm{d}\ell}\mu_1(\ell) = -2(\Gamma_1^2+\mu_1^2(\ell))\mu_1(\ell); \qquad \frac{\mathrm{d}}{\mathrm{d}\ell}\mu_1(\ell) = -\Gamma_1^2\mu_1(\ell); \qquad \frac{\mathrm{d}}{\mathrm{d}\ell}\mu_1(\ell) = -\mu_1(\ell). \tag{59}$$

Whereas the second and third equations are trivial, and are solved by two exponentials, in this limit it is possible to give a simple and analytical solution also to the former:

$$\mu_1(\ell) = \Gamma_1 \frac{1}{\sqrt{2e^{4\Gamma_1^2\ell}-1}}. \tag{60}$$

This results, together with those presented in Sec. 4 highlight the fact that the third generator of the flow is the most efficient from a numerical viewpoint. On the other hand, the first and second generators share several similarities, and given that between the two the second is the most efficient and simplest, we disregard from now on the first generator of the flow.

**Steady-state and long-time properties**

We now present a characterization of the steady state following the guidelines presented in Sec. 2.2. For simplicity, we focus on the charge operator $\hat{c}^\dagger \hat{c}$, that in superfermion notation reads $c^\dagger c$, and introduce its matrix representation during the flow $n(\ell)$:

$$n(\ell) = \begin{pmatrix} n_{11}(\ell) & n_{12}(\ell) \\ n_{21}(\ell) & n_{22}(\ell) \end{pmatrix}; \qquad \text{with} \quad n(\ell = 0) = \begin{pmatrix} 1 & 0 \\ 0 & 0 \end{pmatrix}. \tag{61}$$

Once the flow runs until the final value $\ell \to \infty$, the matrix is a representation of the charge operator in the basis employed in Eq. (41) for the $L$ operator:

$$c^\dagger c = \begin{pmatrix} D^\dagger & \tilde{D} \end{pmatrix} \begin{pmatrix} n_{11}(\ell \to \infty) & n_{12}(\ell \to \infty) \\ n_{21}(\ell \to \infty) & n_{22}(\ell \to \infty) \end{pmatrix} \begin{pmatrix} d \\ \tilde{d}^\dagger \end{pmatrix}. \tag{62}$$

The steady-state charge reads $\langle I | c^\dagger c | \rho_\infty \rangle$ and thanks to Eqs. (42) which characterize the steady-state, and using the anticommutation relations $\{\tilde{D}, \tilde{d}^\dagger\} = 1$ and $\{\tilde{D}, d\} = 0$, we obtain that the charge of the stationary state is $n_{22}(\ell \to \infty)$ (for more details on this calculation, we refer to Appendix B). Thus, our goal is to compute $n(\ell)$ by solving the flow dynamics $\frac{d}{d\ell} n(\ell) = [\eta(\ell), n(\ell)]$.[1]

Using the fact that the third generator admits a fully-analytical expression:

$$\eta(\ell) = \frac{ie^{-\ell}}{\sqrt{(\Gamma_1 + \Gamma_2)^2 - 4\Gamma_1\Gamma_2 e^{-2\ell}}} \begin{pmatrix} 0 & \Gamma_2 \\ \Gamma_1 & 0 \end{pmatrix} = h(\ell) \begin{pmatrix} 0 & \Gamma_2 \\ \Gamma_1 & 0 \end{pmatrix}, \tag{63}$$

we obtain the following set of differential equations:

$$\frac{d}{d\ell} \begin{pmatrix} n_{11}(\ell) & n_{12}(\ell) \\ n_{21}(\ell) & n_{22}(\ell) \end{pmatrix} = h(\ell) \begin{pmatrix} \Gamma_2 n_{21}(\ell) - \Gamma_1 n_{12}(\ell) & \Gamma_2(n_{22}(\ell) - n_{11}(\ell)) \\ -\Gamma_1(n_{22}(\ell) - n_{11}(\ell)) & -\Gamma_2 n_{21}(\ell) + \Gamma_1 n_{12}(\ell) \end{pmatrix}. \tag{64}$$

We do not directly solve these differential equations, but rather compute the matrix $S(\ell) = exp[\int_0^\ell \eta(\ell') d\ell']$; note that path ordering is not necessary because $[\eta(\ell), \eta(\ell')] = 0$ for $\ell \neq \ell'$; thus, one can first compute the integral of the matrix and then takes the exponential. This matrix is useful because $n(\ell) = S(\ell) n(0) S(\ell)^{-1}$; it is not difficult to see that it satisfies the desired differential equation and initial condition. In this way it is possible to reconstruct the density of the quantum dot; we obtain $n_{22}(\ell \to \infty) = \Gamma_2/(\Gamma_1 + \Gamma_2)$, which is the expected result.

We conclude this section with a remark on the time-evolution of the system, and on the way the system approaches stationary properties. Let us write a generic density matrix in the formalism of $\ell \to \infty$ that describes the system at time $t = 0$:

$$|\rho(0)\rangle = \alpha D^\dagger \tilde{d}^\dagger |\rho_\infty\rangle + |\rho_\infty\rangle. \tag{65}$$

We do not introduce unphysical terms that are linear in the fermionic operators and leave aside problems related to the positivity and hermiticity of the represented density matrix. Applying the Liouvillian (41) and recalling the properties (42) we obtain:

$$|\rho(t)\rangle = \alpha e^{-(\Gamma_1 + \Gamma_2)t} D^\dagger \tilde{d}^\dagger |\rho_\infty\rangle + |\rho_\infty\rangle. \tag{66}$$

From this expression we can finally deduce the time-evolution of the expectation value of the charge operator: $n(t) = n_{22}(\ell \to \infty) + \alpha e^{-(\Gamma_1 + \Gamma_2)t} [n_{11}(\ell \to \infty) - n_{22}(\ell \to \infty)]$ From Eq. (64) we obtain $\frac{d}{d\ell}(n_{11} + n_{22}) = 0$ and from the initial condition $n_{11}(0) + n_{22}(0) = 1$ we have:

$$n(t) = \frac{\Gamma_2 + \alpha e^{-(\Gamma_1 + \Gamma_2)t}(\Gamma_1 - \Gamma_2)}{\Gamma_1 + \Gamma_2}. \tag{67}$$

---

[1]The approach that we are using here is slightly different from that described in Sec. 2.2 because we treat the observable as a superoperator acting on the density matrix.

It is relatively easy at this stage to re-express $\alpha$ as a function of the initial charge density: $\alpha = (n(0) - \Gamma_2)\frac{\Gamma_1 + \Gamma_2}{\Gamma_1 - \Gamma_2}$.

# 6 Third example: Dissipative scattering model

We now move to the application of dissipative flow equations to a problem involving many fermionic modes.

## 6.1 The problem

We now discuss a gas of spinless fermions in a $d$-dimensional square box of volume $L^d$ in the presence of a loss mechanism that acts locally; the Lindblad master equation reads:

$$\frac{\mathrm{d}}{\mathrm{d}t}\rho(t) = -\frac{i}{\hbar}[H, \rho(t)] + \int \mathrm{d}\mathbf{x}\, \Gamma(\mathbf{x})\left( \Psi(\mathbf{x})\rho(t)\Psi(\mathbf{x})^\dagger - \frac{1}{2}\left[ \Psi(\mathbf{x})^\dagger \Psi(\mathbf{x})\rho(t) + \rho(t)\Psi(\mathbf{x})^\dagger \Psi(\mathbf{x}) \right] \right). \tag{68}$$

We consider in particular the case of a loss mechanism that is localized around $\mathbf{x} = 0$, and choose the form: $\Gamma(\mathbf{x}) = \gamma\,\delta(\mathbf{x})$. Note that $\gamma$ has the dimension of a volume divided by a time. We furthermore assume that the Hamiltonian is single-particle $H = \sum_{\mathbf{k}} \varepsilon(\mathbf{k})\hat{c}_{\mathbf{k}}^\dagger \hat{c}_{\mathbf{k}}$ and the full dynamics can be written in momentum space as follows:

$$\frac{\mathrm{d}}{\mathrm{d}t}\rho(t) = -\frac{i}{\hbar}\left[ \sum_{\mathbf{k}} \varepsilon(\mathbf{k})\hat{c}_{\mathbf{k}}^\dagger \hat{c}_{\mathbf{k}}, \rho(t) \right] + \frac{\gamma}{L^d} \sum_{\mathbf{k},\mathbf{q}} \left( \hat{c}_{\mathbf{k}}\rho(t)\hat{c}_{\mathbf{q}}^\dagger - \frac{1}{2}\left[ \hat{c}_{\mathbf{k}}^\dagger \hat{c}_{\mathbf{q}}\rho(t) + \rho(t)\hat{c}_{\mathbf{k}}^\dagger \hat{c}_{\mathbf{q}} \right] \right). \tag{69}$$

This master equation is quadratic in the fermionic operators and amenable to the treatment with fermionic superoperators recalled in Sec. 5 and detailed in Appendix B. We now pass to the superoperator representation: $i\hbar\frac{\mathrm{d}}{\mathrm{d}t}|\rho(t)\rangle = L|\rho(t)\rangle$ with

$$L = \sum_{\mathbf{k}} \varepsilon(\mathbf{k})\left( c_{\mathbf{k}}^\dagger c_{\mathbf{k}} + \tilde{c}_{\mathbf{k}}\tilde{c}_{\mathbf{k}}^\dagger \right) - i\frac{\hbar\gamma}{2L^d} \sum_{\mathbf{k},\mathbf{q}} \left( c_{\mathbf{k}}^\dagger c_{\mathbf{q}} - \tilde{c}_{\mathbf{q}}\tilde{c}_{\mathbf{k}}^\dagger \right) - \frac{\hbar\gamma}{L^d} \sum_{\mathbf{k},\mathbf{q}} \tilde{c}_{\mathbf{k}} c_{\mathbf{q}} - \sum_{\mathbf{k}} \left( \varepsilon(\mathbf{k}) + i\frac{\hbar\gamma}{2L^d} \right). \tag{70}$$

In matrix form:

$$M = \begin{pmatrix} H - \frac{i\hbar}{2}\Lambda_1 & 0 \\ -\Lambda_1 & H + \frac{i\hbar}{2}\Lambda_1 \end{pmatrix}, \tag{71}$$

where $H$ is a diagonal matrix with entries $\varepsilon(\mathbf{k})$ and $\Lambda_1$ is a matrix with all matrix elements equal to $\hbar\gamma/L^d$. Since the matrix is block triangular, in order to study its eigenvalues it is sufficient to look for the eigenvalues of the matrix $M' = H - i\hbar\Lambda_1/2$; in the following we will only concentrate on it. This of course prevents a correct reconstruction of the observables, as detailed in Sec. 2.2, but as long as the focus is on the spectrum it is a legitimate restriction.

The problem has been discussed in several articles, see for instance Refs. [53–57]. Here we focus on a simplified situation, that of a one-dimensional setup ($d = 1$) with a linear dispersion relation: $\varepsilon(k) = \hbar v \frac{2\pi}{L} j$, where $v$ is a velocity and $j \in \mathbb{Z}$. In particular, we are interested in an important spectral feature of the model: for $\gamma \geq 4v$, we observe the appearance of a *strongly dissipative state*, namely of an eigenvalue with real part equal to 0 and imaginary part much larger then that of the other eigenvalues:

$$\lambda = \Lambda \tan\left( \frac{\pi}{2}\left( \frac{4v}{\gamma} - 1 \right) \right), \qquad \gamma > 4v\,; \tag{72}$$

where $\Lambda$ is an appropriate energy cutoff. This eigenvalue marks the appearance of a single state where almost all dissipation is concentrated, whereas all other ones are weakly dissipative (see Fig. 5). This separation of time-scales is typical of the quantum Zeno effect. In appendix C we present the analytical solution of this model and the necessary details.

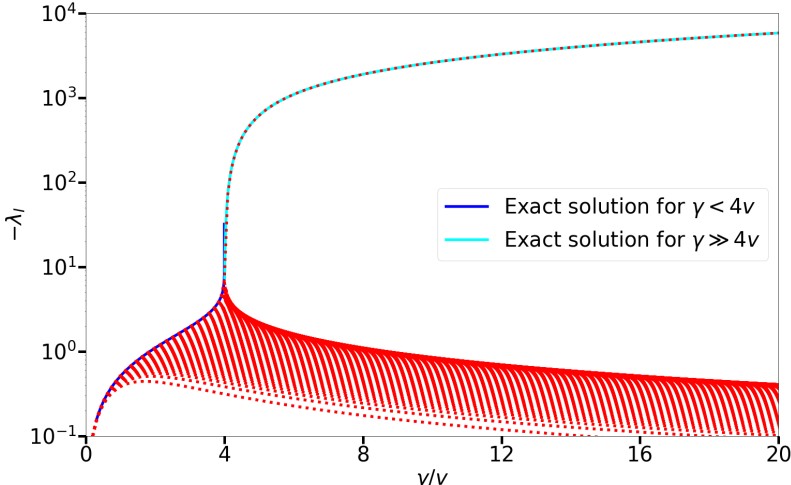

Figure 5: Plot of the imaginary part of the eigenvalues of the dissipative scattering model for $L = 201$ considering 601 states (the parameter $j_\Lambda$ defined in the appendix is set to 300). The plot highlights the emergence of a strongly dissipative state for $\gamma > 4\nu$ which is well described by the formula in (72). For $\gamma < 4\nu$ the formula in 144 in Appendix C describes the imaginary part of the eigenvalue with real part equal to zero that evolves into the strongly dissipative state.

## 6.2 Dissipative flow equations

We will use this example to discuss the dissipative flow equations in momentum space. Following the expression in Eq. (70), we propose the following parametrization for the flow equations:

$$M'(\ell) = \sum_{\mathbf{k}} g_{\mathbf{kk}}(\ell) c_{\mathbf{k}}^\dagger c_{\mathbf{k}} + i \sum_{\mathbf{k} \neq \mathbf{q}} g_{\mathbf{kq}}(\ell) c_{\mathbf{k}}^\dagger c_{\mathbf{q}}, \tag{73}$$

with the following initial conditions:

$$g_{\mathbf{kk}}(0) = \varepsilon(\mathbf{k}) - i\frac{\hbar\gamma}{2L^d}; \qquad g_{\mathbf{kq}}(0) = -\frac{\hbar\gamma}{2L^d}, \text{ for } \mathbf{k} \neq \mathbf{q}. \tag{74}$$

We now divide the super-operator $M'(\ell)$ into a diagonal and an off-diagonal part:

$$D(\ell) = \sum_{\mathbf{k}} g_{\mathbf{kk}}(\ell) c_{\mathbf{k}}^\dagger c_{\mathbf{k}} \qquad V(\ell) = i \sum_{\mathbf{k} \neq \mathbf{q}} g_{\mathbf{kq}}(\ell) c_{\mathbf{k}}^\dagger c_{\mathbf{q}}; \tag{75}$$

and apply the theory that we have developed for the dissipative flow equations.

The flow is characterized by several invariants $I_n$, the first and the second read:

$$I_1 = \sum_{\mathbf{k}} g_{\mathbf{kk}}(\ell); \qquad I_2 = I_2^{\mathrm{diag}}(\ell) + I_2^{\mathrm{off}}(\ell) = \sum_{\mathbf{k}} g_{\mathbf{kk}}^2(\ell) - \sum_{\mathbf{k} \neq \mathbf{q}} g_{\mathbf{kq}}(\ell) g_{\mathbf{qk}}(\ell). \tag{76}$$

**Second generator**

We proceed to the solution of the problem using the second generator of the dynamics:

$$\eta(\ell) = [D(\ell)^\dagger, V(\ell)], \tag{77}$$

whose explicit expression reads (we omit for brevity the dependence on $\ell$):

$$[D(\ell)^\dagger, V(\ell)] = \sum_{\mathbf{k}} \sum_{\mathbf{s} \neq \mathbf{q}} i\, g_{\mathbf{kk}}^* g_{\mathbf{sq}} \left[ c_{\mathbf{k}}^\dagger c_{\mathbf{k}}, c_{\mathbf{s}}^\dagger c_{\mathbf{q}} \right] = \sum_{\mathbf{s} \neq \mathbf{q}} i \left( g_{\mathbf{ss}}^* - g_{\mathbf{qq}}^* \right) g_{\mathbf{sq}}\, c_{\mathbf{s}}^\dagger c_{\mathbf{q}} = \sum_{\mathbf{sq}} \eta_{\mathbf{sq}}\, c_{\mathbf{s}}^\dagger c_{\mathbf{q}}; \tag{78}$$

where we have used the important identity:

$$\left[c_{\mathbf{k}}^{\dagger}c_{\mathbf{t}}, c_{\mathbf{s}}^{\dagger}c_{\mathbf{q}}\right] = \delta_{\mathbf{st}}c_{\mathbf{k}}^{\dagger}c_{\mathbf{q}} - \delta_{\mathbf{kq}}c_{\mathbf{s}}^{\dagger}c_{\mathbf{t}}. \tag{79}$$

We now compute the commutator $[\eta(\ell), M'(\ell)]$ by splitting it into two parts; first the commutator with $D(\ell)$

$$[\eta(\ell), D(\ell)] = \sum_{\mathbf{k}}\sum_{\mathbf{s}\neq\mathbf{q}} g_{\mathbf{kk}}\, \eta_{\mathbf{sq}}\left[c_{\mathbf{s}}^{\dagger}c_{\mathbf{q}}, c_{\mathbf{k}}^{\dagger}c_{\mathbf{k}}\right] =$$
$$= \sum_{\mathbf{k}\neq\mathbf{q}}(g_{\mathbf{qq}}-g_{\mathbf{kk}})\eta_{\mathbf{kq}}c_{\mathbf{k}}^{\dagger}c_{\mathbf{q}} = \sum_{\mathbf{k}\neq\mathbf{q}} i|g_{\mathbf{kk}}-g_{\mathbf{qq}}|^{2}g_{\mathbf{kq}}c_{\mathbf{k}}^{\dagger}c_{\mathbf{q}}; \tag{80}$$

and then the commutator with $V(\ell)$:

$$[\eta(\ell), V(\ell)] = \sum_{\mathbf{k}\neq\mathbf{t}}\sum_{\mathbf{s}\neq\mathbf{q}} ig_{\mathbf{kt}}\, \eta_{\mathbf{sq}}\left[c_{\mathbf{s}}^{\dagger}c_{\mathbf{q}}, c_{\mathbf{k}}^{\dagger}c_{\mathbf{t}}\right] = \sum_{\mathbf{kqs}} i\left(g_{\mathbf{sq}}\eta_{\mathbf{ks}}-g_{\mathbf{ks}}\eta_{\mathbf{sq}}\right)c_{\mathbf{k}}^{\dagger}c_{\mathbf{q}} =$$
$$= -\sum_{\mathbf{ks}} 2(g_{\mathbf{kk}}^{*}-g_{\mathbf{ss}}^{*})\, g_{\mathbf{sk}}\, g_{\mathbf{ks}}c_{\mathbf{k}}^{\dagger}c_{\mathbf{k}} - \sum_{\mathbf{k}\neq\mathbf{q}}\sum_{\mathbf{s}} g_{\mathbf{ks}}\, g_{\mathbf{sq}}\left(g_{\mathbf{qq}}^{*}+g_{\mathbf{kk}}^{*}-2g_{\mathbf{ss}}^{*}\right)c_{\mathbf{k}}^{\dagger}c_{\mathbf{q}}. \tag{81}$$

With this information we can write the flow equations for the coupling constants:

$$\frac{\mathrm{d}}{\mathrm{d}\ell}g_{\mathbf{kk}}(\ell) = -\sum_{\mathbf{s}} 2(g_{\mathbf{kk}}^{*}-g_{\mathbf{ss}}^{*})\, g_{\mathbf{ks}}\, g_{\mathbf{sk}}; \tag{82a}$$

$$\frac{\mathrm{d}}{\mathrm{d}\ell}g_{\mathbf{kq}}(\ell) = -|g_{\mathbf{kk}}-g_{\mathbf{qq}}|^{2}g_{\mathbf{kq}} + i\sum_{\mathbf{ss}} g_{\mathbf{ks}}\, g_{\mathbf{sq}}\left(g_{\mathbf{qq}}^{*}+g_{\mathbf{kk}}^{*}-2g_{\mathbf{ss}}^{*}\right). \tag{82b}$$

**Third generator**

According to the general definition, we propose the following generator of the dynamics:

$$\eta(\ell) = \sum_{\mathbf{k}\neq\mathbf{q}}\frac{g_{\mathbf{kq}}}{g_{\mathbf{kk}}-g_{\mathbf{qq}}}c_{\mathbf{k}}^{\dagger}c_{\mathbf{q}}. \tag{83}$$

In order to compute the commutator $[\eta(\ell), L(\ell)]$, we can reemploy some of the results obtained for the second commutator, and obtain:

$$[\eta(\ell), D(\ell)] = \sum_{\mathbf{k}\neq\mathbf{q}} g_{\mathbf{kq}}c_{\mathbf{k}}^{\dagger}c_{\mathbf{q}}; \tag{84a}$$

$$[\eta(\ell), V(\ell)] = \sum_{\mathbf{ks}} 2\frac{g_{\mathbf{sk}}\, g_{\mathbf{ks}}}{g_{\mathbf{ss}}-g_{\mathbf{kk}}}c_{\mathbf{k}}^{\dagger}c_{\mathbf{k}} + \sum_{\mathbf{k}\neq\mathbf{q}}\sum_{\mathbf{s}} g_{\mathbf{ks}}\, g_{\mathbf{sq}}\frac{2g_{\mathbf{ss}}-g_{\mathbf{qq}}-g_{\mathbf{kk}}}{(g_{\mathbf{ss}}-g_{\mathbf{qq}})(g_{\mathbf{kk}}-g_{\mathbf{ss}})}c_{\mathbf{k}}^{\dagger}c_{\mathbf{q}} + .. \tag{84b}$$

We are now ready to write the flow equations for the third generator:

$$\frac{\mathrm{d}}{\mathrm{d}\ell}g_{\mathbf{kk}}(\ell) = -2\sum_{\mathbf{s}\neq\mathbf{k}}\frac{g_{\mathbf{sk}}\, g_{\mathbf{ks}}}{g_{\mathbf{kk}}-g_{\mathbf{ss}}} \tag{85a}$$

$$\frac{\mathrm{d}}{\mathrm{d}\ell}g_{\mathbf{kq}}(\ell) = -g_{\mathbf{kq}} + i\sum_{\mathbf{s}\neq\mathbf{k},\mathbf{q}} g_{\mathbf{ks}}\, g_{\mathbf{sq}}\frac{2g_{\mathbf{ss}}-g_{\mathbf{qq}}-g_{\mathbf{kk}}}{(g_{\mathbf{ss}}-g_{\mathbf{qq}})(g_{\mathbf{kk}}-g_{\mathbf{ss}})}. \tag{85b}$$

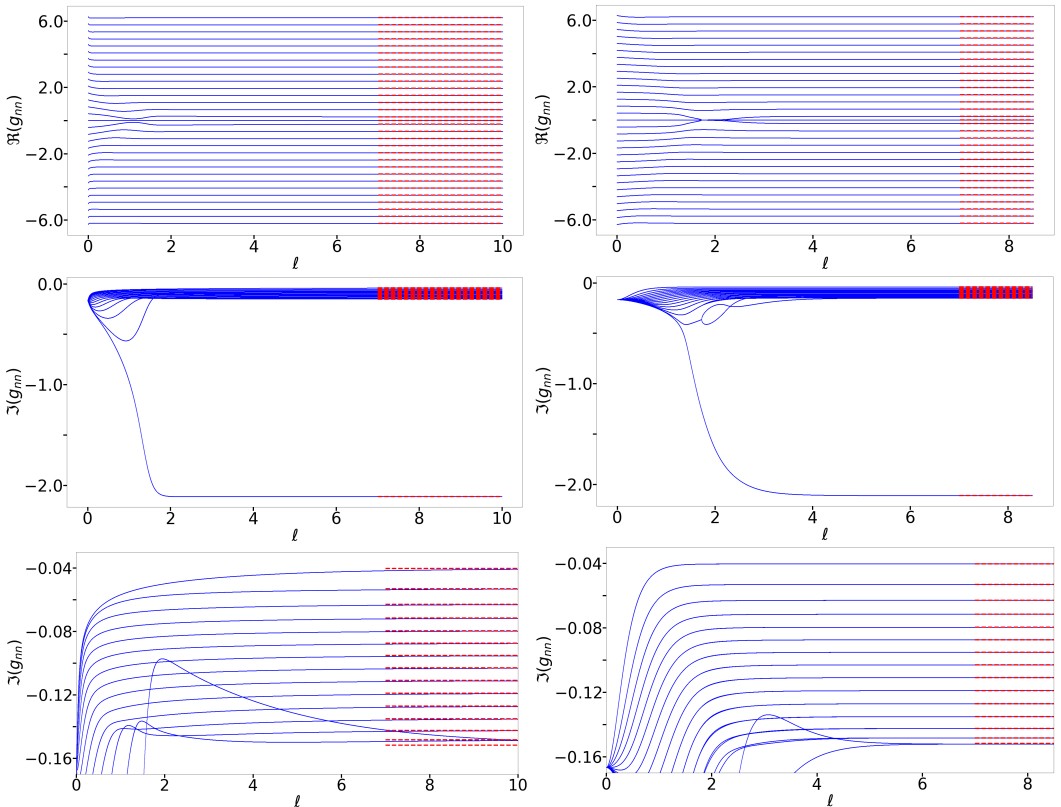

Figure 6: Numerical solutions of the flow equations with the $2^{nd}$ generator in (82) (left column) and with the and $3^{rd}$ generator in (85) (right column). We plot the real (top line) and imaginary (middle line) parts of the diagonal elements $g_{\mathbf{kk}}(\ell)$; the bottom line is a zoom into the imaginary parts with smallest absolute value. The dotted red lines represent the correct values computed with a standard linear algebra package. We observe that the system has reached a diagonal form with a good approximation in both cases, although the bottom left panel shows that the flow of the second generator is not yet at complete convergence.

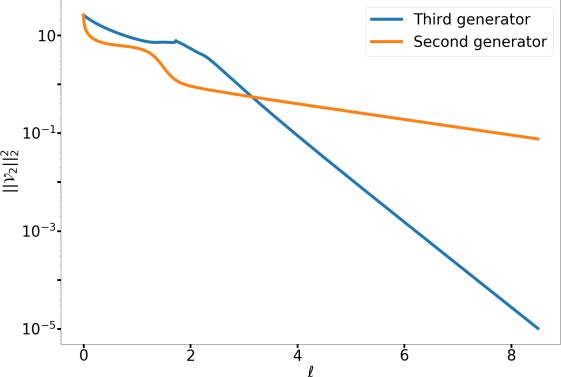

Figure 7: Behaviour during the flow of $\|\mathcal{V}(\ell)\|_2^2$ for the second (light blue) and third (red) generator. In this latter case, we recover the behaviour predicted in formula (35). The second generator instead has a less uniform behaviour, although at long times it also show an exponential law.

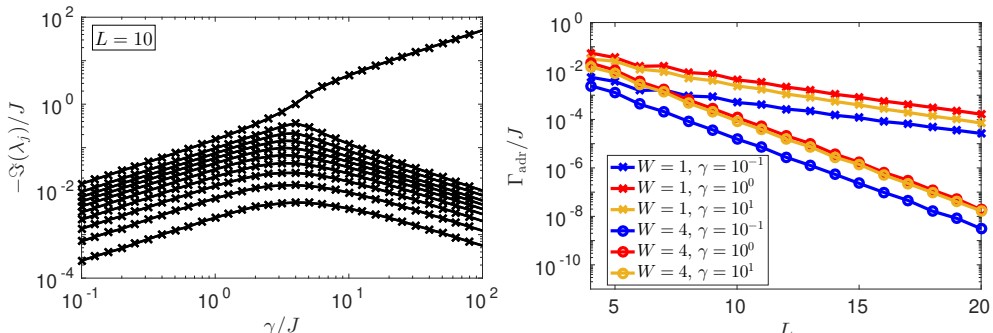

Figure 8: Left: Spectrum of the matrix $M'$ for a lattice of 10 sites and $W = J$ after averaging over $10^3$ disorder realisations. The plot shows the appearance of a strongly dissipative state for $\gamma > 4J$ which is separated by a set of states which are quasi-stationary with lifetime scaling as $\gamma^{-1}$. Right: Size-scaling of the asymptotic decay rate $\Gamma_{\text{adr}}$ for different valus of $\gamma$ and $W$. Each point is obtained by averaging over $10^4$ disorder realizations. In all situations we observe an exponential scaling with the lattice size, $\Gamma_{\text{adr}} \sim e^{-L}$. All results presented in this figure have been obtained with standard diagonalization routines.

**Numerical solutions**

We have solved numerically the flow equations in (82) and in (85). In the following we present the numerical results obtained by solving the flow equations for the dissipative scattering model both with the second and third generator. We set $\gamma = 5v$ and $j_\Lambda = 15$ (so that we consider 31 states in total). The numerical algorithm that has been used to solve the system of coupled ordinary differential equations is a $5^{\text{th}}$ order Runge-Kutta (Butcher's scheme) with a flow step $d\ell = 10^{-4}$ and $N_{step} = 10^5$ for the second generator, whereas $N_{step} = 8.5 \cdot 10^4$ for the third generator.

The results are summarized in Fig. 6. We first observe that both generators let the system converge to a diagonal form. In order to be more quantitative, the discrepancy $\Delta(\ell_{\max})$ for the second generator equals $6.0 \cdot 10^{-3}$, whereas for the third generator is $8.11 \cdot 10^{-11}$. Concerning the conserved quantities, we observe that $\delta I_{31}$ equals $1.8 \cdot 10^{-2}$ for the second flow and $3.8 \cdot 10^{-6}$ for the third one. In Fig. 7 we show the flow of $\|\mathcal{V}(\ell)\|_2^2$ and observe that the third generator produces a more effective approach to zero.

This study corroborates the previous conclusions on the fact that for numerical applications the third generator is more effective than the second one. On the other hand, the dynamics generated by the second generator is more uniform than that of the third generator. In this respect, we envision that the use of the second generator would be more useful in situations where an approximated treatment is necessary (e.g. for interacting non-quadratic systems).

## 7 Fourth example: Disordered dissipative scattering model

In this Section we continue our analysis of dissipative flow equations applied to a many-fermion problem and consider a one-dimensional lattice with a fermionic disordered tight-binding model:

$$\hat{H} = -J \sum_j \left[ \hat{c}_j^\dagger \hat{c}_{j+1} + H.c. \right] + \sum_j h_j \hat{n}_j. \tag{86}$$

Here, $h_j$ takes random values and is uniformly distributed in the range $[-W, W]$. We consider a localized loss source at the center of the lattice ($j = 0$) with loss rate $\gamma$, so that the master

equation reads: $\frac{\mathrm{d}}{\mathrm{d}t}\rho(t) = -\frac{i}{\hbar}[H, \rho(t)] + \gamma\left(\hat{c}_0\rho(t)\hat{c}_0^\dagger - \frac{1}{2}\{\hat{n}_0, \rho(t)\}\right)$. In this example we aim at investigating the interplay between disorder and losses, and at discussing the emerging behaviour of the system as a dissipative insulator.

This analysis can be performed by focusing on the size-scaling of the asymptotic decay rate, namely the longest typical decay time of the system, which is related to the spectrum of the Lindbladian. Similarly to what done in Sec. 6, we rewrite the master equation using the fermionic superoperators (see Appendix B) and link the spectrum of the master equation to the eigenvalues $\lambda_\alpha$ of the matrix $M' = H - i\hbar\Lambda_1/2$, which represents the master equation in this formalism. The asymptotic decay rate reads:

$$\Gamma_{\mathrm{adr}} = \min_\alpha\left(-\Im[\lambda_\alpha]\right). \tag{87}$$

In Fig. 8 we show the asymptotic decay rate of the problem, obtained with exact diagonalization, for several values of the lattice length $L$ and of the decay rate $\gamma$. Its scaling is exponential in the system size: $\Gamma_{\mathrm{adr}} \sim \exp[-L]$ and this behaviour is solely dictated by the presence of disorder. Indeed, we verified that for a clean system the scaling is always algebraic $L^{-\alpha}$. Similarly to the model discussed in Sec. 6, the system also displays a strongly dissipative state for $\gamma > 4J$, both in the clean [54, 56] and in the disordered case (see Fig. 8). The latter result, related to the disordered model, has not been thoroughly discussed yet and deserves further investigation.

## 7.1 Dissipative flow equations

We propose to study this model using flow equations that are formulated in real space:

$$M'(\ell) = \sum_j g_{jj}(\ell)c_j^\dagger c_j + i\sum_{j\neq j'} g_{jj'}(\ell)c_j^\dagger c_{j'}, \tag{88}$$

with initial conditions:

$$g_{jj}(0) = h_j - i\frac{\hbar\gamma}{2}\delta_{j,0}; \qquad g_{j,j'}(0) = \frac{-J}{i}\left(\delta_{j',j+1} + \delta_{j',j-1}\right). \tag{89}$$

The flow equations are not different from those presented in Sec. 6.2 and we limit ourselves here to writing the final results. For the second generator of the flow:

$$\frac{\mathrm{d}}{\mathrm{d}\ell}g_{jj}(\ell) = -\sum_{j'} 2(g_{jj}^* - g_{j'j'}^*)g_{jj'}g_{j'j}; \tag{90a}$$

$$\frac{\mathrm{d}}{\mathrm{d}\ell}g_{jj'}(\ell) = -|g_{jj} - g_{j'j'}|^2 g_{jj'} + i\sum_n g_{jn}g_{nj'}\left(g_{j'j'}^* + g_{jj}^* - 2g_{nn}^*\right). \tag{90b}$$

For the third generator of the flow:

$$\frac{\mathrm{d}}{\mathrm{d}\ell}g_{jj}(\ell) = -2\sum_{j'\neq j}\frac{g_{j'j}g_{jj'}}{g_{jj} - g_{j'j'}}; \tag{91a}$$

$$\frac{\mathrm{d}}{\mathrm{d}\ell}g_{jj'}(\ell) = -g_{jj'} + i\sum_{n\neq j,j'} g_{jn}g_{nj'}\frac{2g_{nn} - g_{j'j'} - g_{jj}}{(g_{nn} - g_{j'j'})(g_{jj} - g_{nn})}. \tag{91b}$$

We present a numerical solution of the flow dynamics; we use a 5-th order Runge-Kutta algorithm (Butcher's scheme) with adaptive flow steps keeping an estimated error below the threshold of $10^{-16}$ according to Butcher's tableau.

In the top panels of Fig. 9 we show the flow of the imaginary part of the diagonal matrix elements $g_{jj}(\ell)$ averaged over $10^4$ disorder realizations obtained with the second and the third

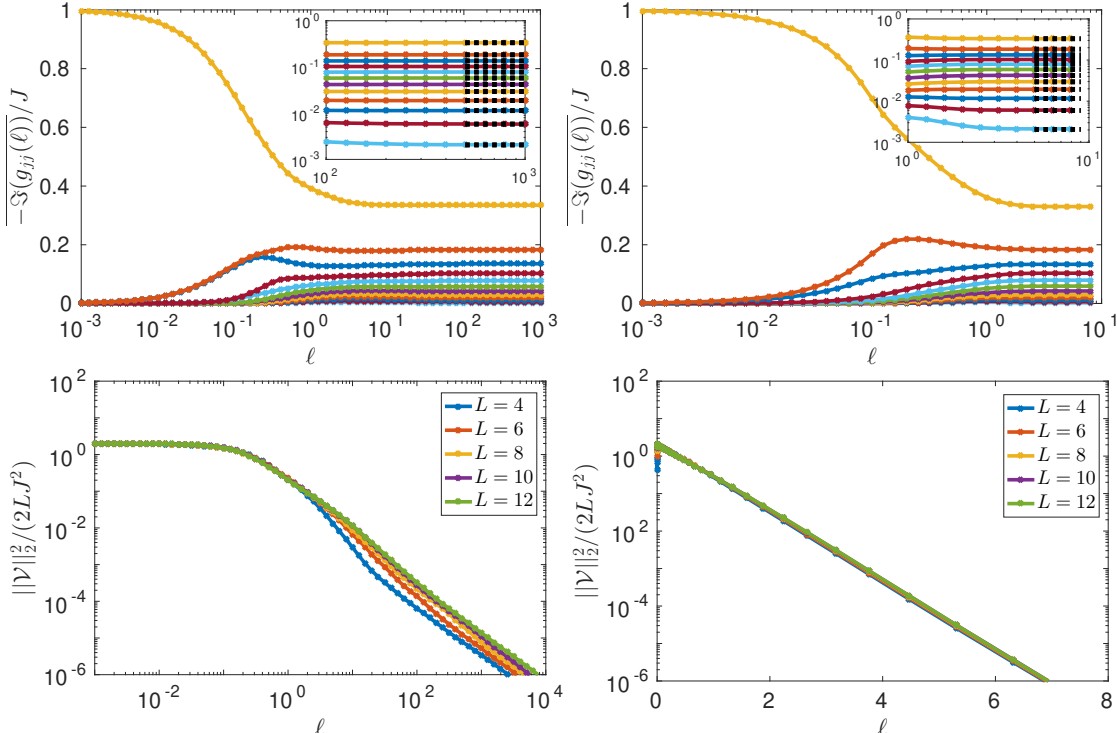

Figure 9: Top: Numerical solutions of the flow equations with the $2^{nd}$ generator in (90) (left) and with the $3^{rd}$ generator in (91) (right). We plot the imaginary part of the diagonal elements $g_{jj}(\ell)$ averaged over 200 (left) and 1000 (right) disorder realisations in the case $L = 12$, $\gamma = 2J$ and $W = 1$. In the insets we highlight a comparison with the expected values computed with standard linear-algebra routines (black dashed lines). Bottom: Behaviour of $\|\mathcal{V}\|_2^2$ with the flow; the law is algebraic for the second generator (left) and exponential for the third one (right).

flow generators. In both cases we see a good convergence to the expected values, computed with standard linear-algebra numerical packages. This convergence is associated to the disappearance during the flow of the off-diagonal part of the matrix. The bottom panels of Fig. 9 show that with the second generator the law is algebraic, whereas with the third generator it is exponential in $\ell$.

# 8 Conclusion

In this paper we have generalized to open quantum systems the flow equations that have originally been developed by Wegner, Głazek and Wilson. Specifically, our work shows how to generalize the flow equations to operators that are not Hermitian, focusing in particular on fermionic Lindblad master equations. Although we did not discuss it explicitly, our results can also be employed for non-Hermitian Hamiltonians and in general for other master equations with time-evolution generators that are local in time and time-independent, such as Redfield master equations [58].

We have described three generators of the flow and have highlighted their peculiarities and strong points. In particular, we have shown that the third generator is the most suited for a numerical use without any physical approximation: its convergence has proven fast and reliable, which is necessary and desirable for implementing a stable and efficient algorithm.

The main perspective of this work is related to the believe that the second generator of the

flow could find a fruitful application in novel approximations schemes aiming at the development of renormalization group-like approaches. We have indeed shown that, in the long flow limit, the off-diagonal matrix elements approach zero with a flow scale that depends on the difference of the two eigenvalues that they connect. This behaviour is reminiscent of decimation schemes proposed for renormalization groups of Hamiltonians, as discussed for instance in Ref. [18]; this property constitute a conceptual asset that is lacking in the case of the third generator. Whereas we have benchmarked our new method with four different examples, in all these cases efficient techniques for the solution of the dynamics exist. On the other hand, the interest of dissipative flow equations might reside in the development of novel truncation and approximation schemes for the treatment of interacting problems in the presence of dissipation, where instead we lack a general and well-established method to use. The second generator constitutes the best candidate for this studies.

## Acknowledgements

We gratefully acknowledge discussions with A. Biella, A. Rosso, C. Mora and T. Maimbourg. F. I. gratefully acknowledges hospitality from LPTMS.

**Funding information** F. I., L. M. and M. S. acknowledge support at different levels from LabEx PALM (ANR-10-LABX-0039-PALM). F. I. acknowledges the financial support of the Brazilian funding agencies National Council for Scientific and Technological Development CNPq (Grant No.308205/2019-7) and FAPERJ (Grant No.E-26/211.318/2019). The support of Paris-Saclay University through an incoming master fellowship is gratefully acknowledged.

## A Dissipative Schrieffer-Wolff transformation

We present a new derivation of the Schrieffer-Wolff transformation for Lindbladian operators (and in general for non-Hermitian linear operators) based on the considerations presented in Ref. [59] for Hamiltonian perturbation theory. The same results have already been derived in Ref. [39] using the resolvant method detailed in Ref. [60].

We consider a quantum system whose dynamics is described by a Lindblad master equation

$$\frac{\mathrm{d}}{\mathrm{d}t}\rho(t) = \mathcal{L}[\rho(t)] \tag{92}$$

and assume that the superoperator $\mathcal{L}$ can be written as the sum of two parts

$$\mathcal{L} = \mathcal{L}_0 + \xi\mathcal{L}_1, \tag{93}$$

where $\xi$ is a dimensionless quantity which plays the role of a perturbative parameter. We assume that $\mathcal{L}_0$ can be easily diagonalized and that its eigenvalues $\lambda_{\alpha i}$ can be grouped into well-separated sets labeled by $\alpha$. Eigenvalues relative to different values of $\alpha$ are very different, but within each set $\alpha$ there is not exact degeneracy, so that an additional index $i$ is necessary. The right eigenvectors $\{|\alpha, v_i\rangle\}_i$ span the subspaces $\mathcal{E}_\alpha^{(0)}$, the left eigenvectors are noted $|\alpha, u_j\rangle$ and the following relations hold:

$$\mathcal{L}_0|\alpha, v_i\rangle = \lambda_{\alpha i}|\alpha, v_i\rangle, \quad \langle\alpha, u_i|\mathcal{L}_0 = \lambda_{\alpha i}\langle\alpha, u_i|, \quad \mathcal{P}_\alpha = \sum_i |\alpha, v_i\rangle\langle\alpha, u_i|, \tag{94}$$

where $\mathcal{P}_\alpha$ is the projector onto the subspace $\mathcal{E}_\alpha^{(0)}$ thanks to the relation $\langle\alpha, u_j|\alpha, v_i\rangle = \delta_{ij}$.

The key idea of a Schrieffer-Wolff transformation is to obtain an effective operator $\mathcal{L}'$ that is block-diagonal with respect to the subspaces $\mathcal{E}_\alpha^{(0)}$ (exactly like $\mathcal{L}_0$ is), and that has the same spectrum of $\mathcal{L}$

$$\mathcal{L}' = \sum_\alpha \mathcal{P}_\alpha \mathcal{L}_{\text{eff}}^\alpha \mathcal{P}_\alpha, \qquad \mathcal{P}_\alpha \mathcal{L}' \mathcal{P}_\beta = \delta_{\alpha\beta} \mathcal{L}_{\text{eff}}^\alpha; \tag{95}$$

but that at the same time takes into account presence of $\xi \mathcal{L}_1$, which in principle is not block-diagonal (if it were, there would be no need for this theory). This process will be implemented by an invertible transformation $\mathcal{Q}$ (invertible transformations do preserve the spectrum), which we write as:

$$\mathcal{Q} = e^\eta, \quad \mathcal{Q}^{-1} = e^{-\eta}; \qquad \mathcal{L}' = \mathcal{Q} \mathcal{L} \mathcal{Q}^{-1}. \tag{96}$$

Using the Taylor series of the matrix exponential, it is possible to give another expression to this latter formula using nested commutators:

$$\mathcal{L}' = \mathcal{L} + \left[ \eta, \mathcal{L} \right] + \frac{1}{2!} \left[ \eta, \left[ \eta, \mathcal{L} \right] \right] + \dots. \tag{97}$$

In order to determine the matrix elements of $\eta$, we will employ a perturbative approach in $\xi$ and expand $\eta$ in powers of $\xi$:

$$\eta = \xi \eta^{(1)} + \xi^2 \eta^{(2)} + \xi^3 \eta^{(3)} + \dots + \xi^n \eta^{(n)} + \dots. \tag{98}$$

We do not include any zero-th order term because in the case $\xi = 0$ the operator is already block diagonal and the invertible transformation should be $\mathcal{Q} = I$, which is what one obtains with $\eta = 0$. At this point we have to consider Eq. (97) and compare terms of the same order in $\xi$. For the left-hand-side of the equation we introduce the notation:

$$\mathcal{L}' = \mathcal{L}^{(0)} + \xi \mathcal{L}^{(1)} + \xi^2 \mathcal{L}^{(2)} + \dots \tag{99}$$

For terms at zero-th order in $\xi$ we obtain $\mathcal{L}^{(0)} = \mathcal{L}_0$; for those proportional to $\xi$ we obtain the interesting equality:

$$\mathcal{L}^{(1)} = \left[ \eta^{(1)}, \mathcal{L}_0 \right] + \mathcal{L}_1. \tag{100}$$

Exploiting the fact that the matrix elements of $\mathcal{L}^{(1)}$ between two different manifolds must be zero, see Eq. (95), one obtains the equation:

$$\langle \alpha, u_j | \eta^{(1)} | \beta, v_i \rangle \left( \lambda_{\beta i} - \lambda_{\alpha j} \right) + \langle \alpha, u_j | \mathcal{L}_1 | \beta, v_i \rangle = 0, \quad \text{for } \alpha \neq \beta. \tag{101}$$

The latter equation determines the matrix elements of $\eta^{(1)}$ between two subspaces with different label $\alpha$:

$$\langle \alpha, u_j | \eta^{(1)} | \beta, v_i \rangle = \frac{\langle \alpha, u_j | \mathcal{L}_1 | \beta, v_i \rangle}{\lambda_{\alpha j} - \lambda_{\beta i}}, \quad \alpha \neq \beta. \tag{102}$$

For $\alpha = \beta$, the matrix element is not unambiguously determined. This ambiguity follows from the fact that once the matrix $\eta$ has been found, it is possible to construct an infinite number of other solutions by simply applying an arbitrary invertible transformation to $\mathcal{Q}$ that does not mix different subspaces. One possibility to remove this uncertainty is to impose that $\eta$ has no matrix elements inside each manifolds: $\mathcal{P}_\alpha \eta \mathcal{P}_\alpha = 0$, $\forall \alpha$. For this reason, we set $\langle \alpha, u_j | \eta^{(1)} | \alpha, v_i \rangle = 0$.

We can now determine the matrix elements of $\mathcal{L}_{\text{eff}}^\alpha$ up to second order in $\xi$. For the zero-th order, we simply obtain $\mathcal{P}_\alpha \mathcal{L}_0 \mathcal{P}_\alpha$. For the first order, instead, unsing Eq. (99) and observing that $\left[ \eta^{(1)}, \mathcal{L}_0 \right]$ has only matrix elements between states with different values of $\alpha$, we obtain: $\mathcal{P}_\alpha \mathcal{L}_1 \mathcal{P}_\alpha$.

For the second order, the key equation is:

$$\mathcal{L}^{(2)} = [\eta^{(1)}, \mathcal{L}_1] + [\eta^{(2)}, \mathcal{L}_0] + \frac{1}{2}[\eta^{(1)}, [\eta^{(1)}, \mathcal{L}_0]], \tag{103}$$

and our goal is to determine $\mathcal{P}_\alpha \mathcal{L}^{(2)} \mathcal{P}_\alpha$. Since $\eta^{(2)}$ has only matrix elements between states with different values of $\alpha$, we know that $\mathcal{P}_\alpha [\eta^{(2)}, \mathcal{L}_0] \mathcal{P}_\alpha = 0$. Moreover, using Eq. (101) one obtains: $[\eta^{(1)}, [\eta^{(1)}, \mathcal{L}_0]] = [\eta^{(1)}, \mathcal{L}^{(1)}] - [\eta^{(1)}, \mathcal{L}_1]$. Since $\mathcal{L}^{(1)}$ has only matrix elements between states with the same $\alpha$, $\mathcal{P}_\alpha [\eta^{(1)}, \mathcal{L}^{(1)}] \mathcal{P}_\alpha = 0$. We are thus left with

$$\mathcal{P}_\alpha \mathcal{L}^{(2)} \mathcal{P}_\alpha = +\frac{1}{2} \mathcal{P}_\alpha [\eta^{(1)}, \mathcal{L}_1] \mathcal{P}_\alpha. \tag{104}$$

And consequently, up to second order we have:

$$\mathcal{L}^\alpha_{\text{eff}} = \mathcal{P}_\alpha \mathcal{L}_0 \mathcal{P}_\alpha + \xi \mathcal{P}_\alpha \mathcal{L}_1 \mathcal{P}_\alpha + \frac{\xi^2}{2} \mathcal{P}_\alpha [\eta^{(1)}, \mathcal{L}_1] \mathcal{P}_\alpha + o(\xi^2). \tag{105}$$

The final step is to evaluate the matrix elements of $[\eta^{(1)}, \mathcal{L}_1]$:

$$\langle \alpha, u_i | [\eta_1, \mathcal{L}_1] | \alpha, v_j \rangle = \sum_{\substack{\beta \neq \alpha \\ k}} \Big( \langle \alpha, u_i | \eta_1 | \beta, v_k \rangle \langle \beta, u_k | \mathcal{L}_1 | \alpha, v_j \rangle - \langle \alpha, u_i | \mathcal{L}_1 | \beta, v_k \rangle \langle \beta, u_k | \eta_1 | \alpha, v_j \rangle \Big) =$$

$$= \sum_{\substack{\beta \neq \alpha \\ k}} \langle \alpha, u_i | \mathcal{L}_1 | \beta, v_k \rangle \langle \beta, u_k | \mathcal{L}_1 | \alpha, v_j \rangle \left( \frac{1}{\lambda_{\alpha i} - \lambda_{\beta k}} + \frac{1}{\lambda_{\alpha j} - \lambda_{\beta k}} \right). \tag{106}$$

Summarizing, in the case in which we have exact degeneracy within each set $\alpha$ (i.e. the subscript $i$ in $\lambda_{\alpha, i}$ is not necessary), we are left with:

$$\mathcal{P}_\alpha \mathcal{L}^{(2)} \mathcal{P}_\alpha = \sum_{\beta \neq \alpha} \mathcal{P}_\alpha \mathcal{L}_1 \mathcal{P}_\beta \mathcal{L}_1 \mathcal{P}_\alpha \left( \frac{1}{\lambda_\alpha - \lambda_\beta} \right). \tag{107}$$

# B Superoperator formalism for fermions

We briefly review the superoperator formalism for fermions (see Refs. [40, 41] for more extensive discussions). It is also worth to mention that this formalism is also connected to the third-quantization formalism presented in Ref. [61], where a real fermion representation is preferred to a complex one. Section B.2 presents some remarks that we did not find explicitly written elsewhere.

## B.1 Generalities

The spirit of the superoperator formalism is to represent density matrices $\rho$ in a space $\mathcal{H} \otimes \tilde{\mathcal{H}}$, where $\mathcal{H}$ is the Hilbert space associated to the physical system (it could be a Fock space), whereas $\tilde{\mathcal{H}}$ is a space that is isomorphic to it. We introduce the basis $\{|n\rangle\}_n$ for $\mathcal{H}$ and the basis $\{|\tilde{n}\rangle\}_n$ for $\tilde{\mathcal{H}}$ and the isomorphism $U : \mathcal{H} \to \tilde{\mathcal{H}}$ that maps $|n\rangle \to |\tilde{n}\rangle$.

With this notation we can introduce the *left vacuum vector*:

$$|I\rangle = \sum_n |n\rangle |\tilde{n}\rangle \in \mathcal{H} \otimes \tilde{\mathcal{H}}, \tag{108}$$

and the representation of the density matrix $\rho = \sum_{nm} \rho_{nm} |n\rangle \langle m|$ as a vector of this space:

$$|\rho\rangle = \sum_{nm} \rho_{nm} |n\rangle |\tilde{m}\rangle = \rho \otimes \hat{I} |I\rangle. \tag{109}$$

The normalization of the density matrix $\text{tr}[\rho] = 1$ reads $\langle I|\rho\rangle = 1$, whereas the expectation value of an observable $\hat{A}$, defined as $\langle A\rangle = \text{tr}[\rho\hat{A}]$, reads $\langle A\rangle = \langle I|\hat{A}\otimes\hat{I}|\rho\rangle$.

We now consider the case where $\mathcal{H}$ is a fermionic Fock space, with $L$ associated fermionic operators $\hat{c}_m$ (without loss of generality, we do not consider explicitly spin) satisfying canonic anticommutation relations:

$$\{\hat{c}_m,\hat{c}_n\} = 0, \quad \{\hat{c}_m^\dagger,\hat{c}_n^\dagger\} = 0, \quad \{\hat{c}_m,\hat{c}_n^\dagger\} = \delta_{mn}. \tag{110}$$

In this case it is customary to define the left vacuum state in a slightly different way with respect to Eq. (108), and namely:

$$|I\rangle = \sum_{n_1,n_2\ldots n_L} (i)^{n_1+n_2+\ldots+n_L}|n_1,n_2,\ldots n_L\rangle|\widetilde{n_1,n_2,\ldots n_L}\rangle. \tag{111}$$

The fermionic superoperators for the space $\mathcal{H}\otimes\tilde{\mathcal{H}}$ are defined as follows:

$$c_m = \hat{c}_m\otimes I; \quad c_m^\dagger = \hat{c}_m^\dagger\otimes I; \quad \tilde{c}_m = (-1)^{\hat{N}}\otimes\hat{c}_m; \quad \tilde{c}_m^\dagger = (-1)^{\hat{N}}\otimes\hat{c}_m^\dagger; \tag{112}$$

the action of the $\hat{c}_m$ on $\tilde{\mathcal{H}}$ is obtained through the isomorphism $U$, and $\hat{N} = \sum_m c_m^\dagger c_m$. The new operators thus satisfy canonical anticommutation relations:

$$\{c_m,c_n\} = 0, \quad \{c_m^\dagger,c_n^\dagger\} = 0, \quad \{c_m,c_n^\dagger\} = \delta_{mn}; \tag{113a}$$

$$\{\tilde{c}_m,\tilde{c}_n\} = 0, \quad \{\tilde{c}_m^\dagger,\tilde{c}_n^\dagger\} = 0, \quad \{\tilde{c}_m,\tilde{c}_n^\dagger\} = \delta_{mn}; \tag{113b}$$

$$\{c_m,\tilde{c}_n\} = 0, \quad \{c_m^\dagger,\tilde{c}_n^\dagger\} = 0, \quad \{c_m,\tilde{c}_n^\dagger\} = 0; \quad \{\tilde{c}_m,c_n^\dagger\} = 0. \tag{113c}$$

As a consequence, we can think of $\mathcal{H}\otimes\tilde{\mathcal{H}}$ as an enlarged Fock space with $2L$ anticommuting modes. Once applied onto the left vacuum state (111), the $c_m$ and $\tilde{c}_m$ operators satisfy the fundamental *tilde conjugation rules*, that are crucial in all subsequent calculations:

$$c_m|I\rangle = -i\tilde{c}_m^\dagger|I\rangle, \qquad c_m^\dagger|I\rangle = -i\tilde{c}_m|I\rangle. \tag{114}$$

The definitions of the $\tilde{c}_m$ operators in Eq. (112) are crucial for ensuring the anticommuting properties in Eq. (113c). In principle they are a choice, but they are highly recommended because they allow to treat $c_m$ and $\tilde{c}_m$ on the same footing.

## B.2 Quadratic Lindblad master equations

**Matrix representation based on superoperators**

We now investigate the writing of a Lindblad master equation (2.1) in superoperator representation assuming that the Hamiltonian is quadratic in the fermionic operators and that the jump operators are linear:

$$\hat{H} = \sum_{mn} h_{mn}\hat{c}_m^\dagger\hat{c}_n, \qquad \hat{L}_\alpha = \sum_m \left(l_{1\alpha m}\hat{c}_m + l_{2\alpha m}\hat{c}_m^\dagger\right). \tag{115}$$

Before continuing, we make the following physical assumption: we are only interested in the study of jump operators that either inject particles in the system or take them out. This means that for a fixed $\alpha$, either the $l_{1\alpha m}$ or the $l_{2\alpha m}$ are all zeros. As a consequence, it will always be true that $l_{1\alpha m}l_{2\alpha n} = 0$. For later convenience, let us write:

$$\sum_\alpha \hat{L}_\alpha^\dagger\hat{L}_\alpha = \sum_{mn}\sum_\alpha \left(l_{1\alpha m}^*\hat{c}_m^\dagger + l_{2\alpha m}^*\hat{c}_m\right)\left(l_{1\alpha n}\hat{c}_n + l_{2\alpha n}\hat{c}_n^\dagger\right) =$$

$$= \sum_{mn}\sum_\alpha \left(l_{1\alpha m}^*l_{1\alpha n} - l_{2\alpha m}l_{2\alpha n}^*\right)\hat{c}_m^\dagger\hat{c}_n + \sum_{m\alpha} l_{2\alpha m}^*l_{2\alpha m} =$$

$$= \sum_{mn}\left(\Lambda_{1mn} - \Lambda_{2mn}\right)\hat{c}_m^\dagger\hat{c}_n + \sum_m \Lambda_{2mm}. \tag{116}$$

In the last line we have defined the Hermitian matrices $\Lambda_1$ and $\Lambda_2$ with matrix elements $\Lambda_{1mn} = \sum_\alpha l^*_{\alpha m} l_{1\alpha n}$ and $\Lambda_{2mn} = \sum_\alpha l_{\alpha m} l^*_{1\alpha n}$.

It is customary to represent the operator $i\hbar\mathcal{L}$ instead of $\mathcal{L}$ for its formal similarity with the Schrödinger equation for pure states. Since it is a linear operator, in the superoperator representation it will have quadratic matrix form $i\hbar\mathcal{L}[\rho] \to L|\rho\rangle$:

$$L = \sum_{mn}\left(h_{mn} - \frac{i\hbar}{2}(\Lambda_{1mn} - \Lambda_{2mn})\right)c^\dagger_m c_n + \left(-h_{mn} - \frac{i\hbar}{2}(\Lambda_{1mn} - \Lambda_{2mn})\right)\tilde{c}^\dagger_n \tilde{c}_m +$$
$$+ i(-i)\hbar\left(\Lambda^*_{1mn} c_m \tilde{c}_n + \Lambda_{2mn} c^\dagger_m \tilde{c}^\dagger_n\right) - i\hbar\sum_m \Lambda_{2mm}. \tag{117}$$

In order to obtain this expression, we have directly promoted every original operator $\hat{c}^{(\dagger)}_m$ to an operator $c^{(\dagger)}_m$ acting on $\mathcal{H} \otimes \tilde{\mathcal{H}}$. Subsequently, we have used the fact that the density matrix $\rho$ commutes with the parity operator $\hat{P} = (-1)^{\sum \hat{c}^\dagger_m \hat{c}_m}$ (we also say that it is an even operator) and thus that it commutes with every operator $\tilde{c}^{(\dagger)}_m$. With these simple rules, every term can be readily obtained. For instance, for what concerns the Hamiltonian dynamics $\hat{H}\rho - \rho\hat{H}$, the first term is easily recast in the superoperator language: $\hat{H}\rho|I\rangle = \sum_{mn} h_{mn} c^\dagger_m c_n|\rho\rangle$; the second instead requires some manipulation:

$$\rho\hat{H}|I\rangle = \rho\sum_{mn} h_{mn} c^\dagger_m c_n|I\rangle = -i\rho\sum_{mn} h_{mn} c^\dagger_m \tilde{c}^\dagger_n|I\rangle = i\sum_{mn} h_{mn} \tilde{c}^\dagger_n \rho c^\dagger_m|I\rangle = i(-i)\sum_{mn} h_{mn} \tilde{c}^\dagger_n \tilde{c}_m|\rho\rangle. \tag{118}$$

Let us now attempt to put the operator $L$ in Eq. (117) in matrix form:

$$L = \begin{pmatrix} c^\dagger_1 & \cdots & c^\dagger_L & \tilde{c}_1 & \cdots & \tilde{c}_L \end{pmatrix} \begin{pmatrix} & & \\ & M & \\ & & \end{pmatrix} \begin{pmatrix} c_1 \\ \vdots \\ c_L \\ \tilde{c}^\dagger_1 \\ \vdots \\ c^\dagger_L \end{pmatrix} + K, \tag{119}$$

where $M$ is a $2L \times 2L$ complex matrix and $K$ is a complex constant. We need to rewrite Eq. (117) as follows:

$$L = \sum_{mn}\left(h_{mn} - \frac{i\hbar}{2}(\Lambda_{1mn} - \Lambda_{2mn})\right)c^\dagger_m c_n + \left(h_{mn} + \frac{i\hbar}{2}(\Lambda_{1mn} - \Lambda_{2mn})\right)\tilde{c}_m \tilde{c}^\dagger_n +$$
$$+ \hbar\left(-\Lambda^*_{1nm} \tilde{c}_m c_n + \Lambda^*_{2mn} c^\dagger_m \tilde{c}^\dagger_n\right) +$$
$$- \sum_m\left(h_{mm} + \frac{i\hbar}{2}(\Lambda_{1mm} - \Lambda_{2mm})\right) - i\hbar\sum_m \Lambda_{2mm}, \tag{120}$$

so that the matrix $M$ has the following block-diagonal form:

$$M = \begin{pmatrix} H - \frac{i\hbar}{2}(\Lambda_1 - \Lambda_2) & \hbar\Lambda^*_2 \\ -\hbar\Lambda_1 & H + \frac{i\hbar}{2}(\Lambda_1 - \Lambda_2) \end{pmatrix}, \tag{121}$$

and the matrices $H$, $\Lambda_1$ and $\Lambda_2$ are Hermitian; moreover

$$K = -\text{tr}[h] + i\frac{\hbar}{2}\text{tr}[\Lambda_1 + \Lambda_2]. \tag{122}$$

We observe that the expression for $L$ derived in the previous equations is a generalisation of Eq. (40) presented in the main text for a dissipative fermionic mode.

**Diagonalization of a quadratic Lindblad master equation**

The matrix $M$ in Eq. (121) satisfies a strong symmetry:

$$M = \Sigma_1 M^\dagger \Sigma_1, \qquad \Sigma_1 = \begin{pmatrix} 0 & I \\ I & 0 \end{pmatrix}, \tag{123}$$

where $I$ is the identity; this is a generic result that is true for any matrix of the form

$$\begin{pmatrix} A & B \\ C & A^\dagger \end{pmatrix}, \tag{124}$$

provided $B$ and $C$ are hermitian. As a consequence, $M$ and $M^\dagger$ have the same spectrum; indeed, if we look at the characteristic polynomial:

$$p_M(\lambda) = \det(M - \lambda I) = \det\left(\Sigma_1 M^\dagger \Sigma_1 - \lambda I\right) =$$
$$= \det\left(\Sigma_1 \left(M^\dagger - \lambda I\right) \Sigma_1\right) = \det\left(M^\dagger - \lambda I\right) = p_{M^\dagger}(\lambda). \tag{125}$$

Since the eigenvalues of $M^\dagger$ are the complex conjugates of those of $M$, we obtain that if $\lambda$ is an eigenvalue of $M$, then this is true also for $\lambda^*$. This means in particular that either the eigenvalues are complex and come in pairs, or they are reals.

We now make the assumption that the Jordan canonical form of the matrix $M$ does not contain any nilpotent part; we are not aware of any physical problem in quantum physics where this matematical object played a role. For this reason, we assume that there is an invertible transformation $S$ that puts $M$ in diagonal form:

$$M = S^{-1} D S, \quad D = \begin{pmatrix} \lambda_1 & & & & & \\ & \ddots & & & & \\ & & \lambda_L & & & \\ & & & \lambda_1^* & & \\ & & & & \ddots & \\ & & & & & \lambda_L^* \end{pmatrix}. \tag{126}$$

Thanks to this, we can write:

$$L = \sum_m \left(\lambda_m D_m^\dagger d_m + \lambda_m^* \tilde{D}_m \tilde{d}_m^\dagger\right) + K, \tag{127}$$

where the operators are defined through the matrix elements of $S$ and $S^{-1}$:

$$\begin{pmatrix} d_1 \\ \vdots \\ d_L \\ \tilde{d}_1^\dagger \\ \vdots \\ d_L^\dagger \end{pmatrix} = S \begin{pmatrix} c_1 \\ \vdots \\ c_L \\ \tilde{c}_1^\dagger \\ \vdots \\ c_L^\dagger \end{pmatrix}; \qquad \begin{pmatrix} D_1^\dagger & \cdots & D_L^\dagger & \tilde{D}_1 & \cdots & \tilde{D}_L \end{pmatrix} = \begin{pmatrix} c_1^\dagger & \cdots & c_L^\dagger & \tilde{c}_1 & \cdots & \tilde{c}_L \end{pmatrix} S^{-1}. \tag{128}$$

It is important to stress that $d_m$ and $D_m^\dagger$ are not Hermitian conjugated operators, similarly for $\tilde{D}$ and $\tilde{d}^\dagger$. Yet, it is possible to show that the operators satisfy canonical anticommutation relations. Obvious results:

$$\{d_m, d_n\} = 0, \quad \{\tilde{d}_m^\dagger, \tilde{d}_n^\dagger\} = 0, \quad \{d_m, \tilde{d}_n^\dagger\} = 0; \tag{129a}$$
$$\{D_m^\dagger, D_n^\dagger\} = 0, \quad \{\tilde{D}_m, \tilde{D}_n\} = 0, \quad \{D_m^\dagger, \tilde{D}_n\} = 0. \tag{129b}$$

The less obvious results $\{d_m, D_{m'}^{\dagger}\} = \delta_{mm'}$ and $\{\tilde{d}_m^{\dagger}, \tilde{D}_{m'}\} = \delta_{mm'}$, follow from the judicious application of the definitions (128). With similar reasonings it is possible to observe that the tilde conjugation rules are satisfied:

$$d_m|I\rangle = -i\tilde{d}_m^{\dagger}|I\rangle, \qquad D_m^{\dagger}|I\rangle = -i\tilde{D}_m|I\rangle. \tag{130}$$

It is interesting to observe that when $\Lambda_2$ is a real matrix, in order to extract the spectrum $\{\lambda\}$ it is not necessary to diagonalize the full matrix $M$. Indeed, we can rewrite the matrix $M$ in Eq. (121) in the following compact way:

$$M = H \otimes \mathbb{I}_2 + -\frac{i\hbar}{2}(\Lambda_1 - \Lambda_2) \otimes \sigma_z + \frac{\hbar}{2}(\Lambda_2^* - \Lambda_1) \otimes \sigma_x + \frac{i\hbar}{2}(\Lambda_2^* + \Lambda_1) \otimes \sigma_y. \tag{131}$$

We now propose the following unitary transformation: $\sigma_x \to -\sigma_y$, $\sigma_y \to -\sigma_z$, $\sigma_z \to \sigma_x$ and obtain the following matrix representation:

$$M = \begin{pmatrix} H - \frac{i\hbar}{2}(\Lambda_2^* + \Lambda_1) & +\frac{i\hbar}{2}(\Lambda_2^* - \Lambda_2) \\ -\frac{i\hbar}{2}(\Lambda_2^* - \Lambda_2) & H + \frac{i\hbar}{2}(\Lambda_2^* + \Lambda_1) \end{pmatrix}. \tag{132}$$

If $\Lambda_2$ is a real matrix, $M$ becomes a block-diagonal matrix and thus, in order to study the spectrum of $M$, it is sufficient to study the spectrum of $H \pm \frac{i\hbar}{2}(\Lambda_1 + \Lambda_2)$. We exploit this possibility in Secs. 6 and 7, where $\Lambda_2 = 0$.

**Back to a fermionic master equation**

We conclude this section with a discussion of the physical meaning of Eq. (127). First of all, by exploiting the canonical anticommutation relations in (129), we rewrite it as:

$$L = \sum_m \left(\lambda_m D_m^{\dagger} d_m - \lambda_m^* \tilde{d}_m^{\dagger} \tilde{D}_m\right). \tag{133}$$

This equation now completely resembles Eq. (41) presented in the main text when dealing with a single fermionic mode in the presence of losses and gain. Diagonalizing our master equation is equivalent to turning it into a form where it looks like a system of single fermionic modes independently coupled to independent sources of particle losses and gain. Thus, the imaginary part of each eigenvalue $\Im[\lambda_\alpha]$ gives us the typical time scales of the decays of the normal modes of the problem. All considerations presented in the main text concerning eigenoperators of the dynamics apply here in the multi-mode case. We can also write that after diagonalisation the original master equation reads:

$$\frac{\mathrm{d}}{\mathrm{d}t}\rho(t) = \sum_\alpha -\frac{i\varepsilon_\alpha}{\hbar}\left[\hat{d}_\alpha^{\dagger}\hat{d}_\alpha, \rho(t)\right] + \Gamma_{\alpha,1}\left(\hat{d}_\alpha\rho(t)\hat{d}_\alpha^{\dagger} - \frac{1}{2}\{\hat{d}_\alpha^{\dagger}\hat{d}_\alpha, \rho(t)\}\right) +$$
$$+ \Gamma_{\alpha,2}\left(\hat{d}_\alpha^{\dagger}\rho(t)\hat{d}_\alpha - \frac{1}{2}\{\hat{d}_\alpha\hat{d}_\alpha^{\dagger}, \rho(t)\}\right), \tag{134}$$

with the operators $\hat{d}_\alpha$ satisfying canonical anticommutation relations.

## C  Dissipative scattering model

In this appendix we derive some exact results and present some considerations for the dissipative scattering model that is discussed in Sec. 6.

## C.1  General considerations

We search the eigenvalues $\lambda$ and eigenvectors $\vec{v}$ of the matrix $M' = H - \frac{i\hbar}{2}\Lambda_1$ where $H$ is a diagonal matrix with entries $\varepsilon(\mathbf{k})$ and $\Lambda_1$ is a matrix with all entries equal to $\gamma/L^d$. We note $v_{\mathbf{k}}$ the entries of $\vec{v}$ and the secular equation reads:

$$\varepsilon(\mathbf{k})v_{\mathbf{k}} - i\frac{\hbar\gamma}{2L^d}\sum_{\mathbf{q}} v_{\mathbf{q}} = \lambda v_{\mathbf{k}}, \qquad \lambda, v_{\mathbf{k}} \in \mathbb{C}. \tag{135}$$

With straightforward manipulations we obtain:

$$v_{\mathbf{k}} = -i\frac{\hbar\gamma}{2L^d}\frac{1}{\lambda - \varepsilon(\mathbf{k})}\sum_{\mathbf{q}} v_{\mathbf{q}} \quad\Rightarrow\quad \sum_{\mathbf{k}}\frac{1}{\lambda - \varepsilon(\mathbf{k})} = i\frac{2L^d}{\hbar\gamma}. \tag{136}$$

This latter equation, can be reformulated as one equation for the real part and one for the imaginary part:

$$\sum_{\mathbf{k}}\frac{\Re[\lambda] - \varepsilon(\mathbf{k})}{|\lambda - \varepsilon(\mathbf{k})|^2} = 0; \qquad -\sum_{\mathbf{k}}\frac{\Im[\lambda]}{|\lambda - \varepsilon(\mathbf{k})|^2} = \frac{2L^d}{\hbar\gamma}. \tag{137}$$

In the following we will discuss some aspects of these eigenvalue equations for a specific form of the energy dispersion relation, $\varepsilon(\mathbf{k})$.

## C.2  The case of a one-dimensional system with a linear spectrum

As anticipated in the main text, we consider a one-dimensional system ($d = 1$) with energies $\varepsilon(k) = \hbar v \frac{2\pi}{L} j$, where $v$ is a velocity and $j \in \mathbb{Z}$; sums over $\mathbf{k}$ are converted into sums over $j$ and for brevity we also introduce $\epsilon_0 = \hbar v \frac{2\pi}{L}$. Let us begin by considering Eq. (137) and let us show that when $\lambda$ is purely imaginary (we thus take $\Re[\lambda] = 0$ and parametrize it as $\lambda = i\lambda_I$) it satisfies the first constraint. We fix a high energy cutoff $\Lambda = \epsilon_0 j_\Lambda$ with $j_\Lambda \gg 1$ and such that $-j_\Lambda < j < j_\Lambda$ and write that:

$$\sum_{\mathbf{k}}\frac{\varepsilon(\mathbf{k})}{|\lambda - \varepsilon(\mathbf{k})|^2} = \lim_{j_\Lambda \to \infty}\sum_{j=1}^{j_\Lambda}\left(\frac{j\epsilon_0}{|i\lambda_I - j\epsilon_0|} + \frac{-j\epsilon_0}{|i\lambda_I + j\epsilon_0|}\right) = 0. \tag{138}$$

It is important to consider the cutoff otherwise one would obtain that every $\lambda$ of the form $\lambda = \varepsilon(\mathbf{k}) + i\lambda_I$ would satisfy the first constraint.

We continue with the second equation:

$$\frac{1}{\lambda_I} + \sum_{j=1}^{j_\Lambda}\frac{2\lambda_I}{j^2\epsilon_0^2 + \lambda_I^2} = -\frac{2L}{\hbar\gamma}. \tag{139}$$

The series can be analytically evaluated:

$$\frac{1}{\lambda_I} - \frac{1}{\lambda_I} + \frac{\pi}{\epsilon_0}\coth\left(\frac{\pi\lambda_I}{\epsilon_0}\right) - \frac{i}{\epsilon_0}\left[\psi\left(j_\Lambda - i\frac{\lambda_I}{\epsilon_0}\right) - \psi\left(j_\Lambda + i\frac{\lambda_I}{\epsilon_0}\right)\right] = -\frac{2L}{\hbar\gamma}; \tag{140}$$

where $\psi(z)$ is the Digamma function. We now consider the large band-width limit, with $|j_\Lambda \pm i\lambda_I/\epsilon_0| \gg 1$, so that the following asymptotic expansion can be used:

$$\psi\left(j_\Lambda - i\frac{\lambda_I}{\epsilon_0}\right) - \psi\left(j_\Lambda + i\frac{\lambda_I}{\epsilon_0}\right) \sim \log\left(\frac{j_\Lambda - i\frac{\lambda_I}{\epsilon_0}}{j_\Lambda + i\frac{\lambda_I}{\epsilon_0}}\right) = -2i\arctan\left(\frac{\lambda_I}{j_\Lambda\epsilon_0}\right). \tag{141}$$

Note that by definition $j_\Lambda \pm i\lambda_I/\epsilon_0$ cannot lie on the negative real axis, where the expansion would be problematic. Concluding, we obtain the following equation for the eigenvalue:

$$\frac{\pi}{\epsilon_0}\coth\left(\frac{\pi\lambda_I}{\epsilon_0}\right) - \frac{2}{\epsilon_0}\arctan\left(\frac{\lambda_I}{j_\Lambda\epsilon_0}\right) = -\frac{2L}{\hbar\gamma}. \tag{142}$$

The equation can be simplified by considering that $j_\Lambda$ is larger than $\Lambda_I/\epsilon_0$, so that we can approximate $\arctan x \sim x$. By introducing the variable $x = \pi\lambda_I/\epsilon_0$, the equation reads:

$$\coth(x) = -\frac{4v}{\gamma} + \frac{2}{\pi}\arctan\left(\frac{x}{\pi j_\Lambda}\right). \tag{143}$$

Although the equations has formally two solutions, for physical reasons we only retain the negative one. The peculiar property of this equation results from the fact that $\coth(x)$ is always smaller than $-1$ for $x < 0$. Thus, we can identify three regimes depending on whether $\gamma/v$ is smaller than 4, larger than 4 or approximately 4. We discuss analytically two of them, whose results can be compared with exact numerics in Fig. 5.

**The case $\gamma/v \ll 4$**

In this case $4v/\gamma \gg 1$ and the solution must satisfy $|x| \ll 1$. The correction due to the $\arctan(x/(\pi j_\Lambda))$, that in this case can be approximated by $x/(\pi j_\Lambda)$, is negligible and can be safely disregarded. The eigenvalue reads:

$$\lambda_I = -\frac{2\hbar v}{L}\coth^{-1}\left(\frac{4v}{\gamma}\right) = -\frac{\hbar v}{L}\log\left(\frac{4v/\gamma + 1}{4v/\gamma - 1}\right). \tag{144}$$

The formula is well-defined only for $\gamma < 4v$ and displays a divergence for $\gamma \to 4v^-$ that we do not consider as physical because it is not in the regime of validity of the approximations.

In the deep perturbative limit $\gamma \ll v$ the result gives:

$$\lambda_I = -\frac{\hbar v}{L}\log\left(\frac{1 + \gamma/(4v)}{1 - \gamma/(4v)}\right) \simeq -\frac{\hbar v}{L}\log\left(1 + \frac{2\gamma}{4v}\right) \simeq -\frac{\hbar\gamma}{2L}. \tag{145}$$

**The case $\gamma/v \gg 4$**

In this case $4v/\gamma \ll 1$ and the solution must satisfy $|x| \gg 1$. In this region, we can approximate $\coth(x) \sim -1$ and obtain:

$$\lambda_I = \Lambda\tan\left(\frac{\pi}{2}\left(\frac{4v}{\gamma} - 1\right)\right). \tag{146}$$

We thus obtain that the result is proportional to the band edge, with limiting value for $\gamma/v \to \infty$ equal to $-\infty$.

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
