# Peer review of "Dissipative flow equations"

_SciPost Physics, doi:SciPost Phys. 9, 091 (2020)_

## Round 1 · Referee Report · François Damanet (Referee 1) · 2020-8-31

Strengths

  • The method is generally well described and detailed in the main text, and several appendices complete it very well.
  • The manuscript presents 4 applications of the method in different contexts.

Weaknesses

  • The language is sometimes difficult to understand for a physicist more familiar with standard open quantum system methods. The connection could be slightly clearer.
  • The method derived in the manuscript does not provide at this point a clear advantage compared to other existing methods, but has potential for future works.

Report

The manuscript presents the generalization of the flow equations to open quantum systems described via standard Markovian Lindblad master equation. After a well-motivated introduction, the authors describe the spirit of flow equations, define and derive the corresponding related quantities and finally apply the methods in four different contexts.

While the method derived and used in the manuscript does not provide an advantage compared to other existing methods to solve the investigated problems, the authors mention possible future interesting perspectives for future works.

I would thus recommend the manuscript for publication in SciPost, as I believe this manuscript would be a solid basis for future applications of the method, provided the authors address the following (small) comments/suggestions.

Requested changes

  1. In the presentation of the different generators $\eta$, the authors rely a few times on a perturbative treatment involving some parameter $\xi$. In order to make a clearer connection to concrete physical problems, the authors may want to give a concrete physical interpretation of $\xi$ in the case of one or two simple examples.

  2. In the first example of section 4, I find that the explanation of the procedure to obtain the results plotted in Fig. 1-4 could be clearer.

  3. What could correspond to the non-Hermitian matrix A (a Liouvillian or a non-Hermitian Hamiltonian ?) ?
  4. What 'flow equations' presented above are concretely solved ? Some references to the equations could be useful.
  5. the 'flow step' in Section 4 is denoted by $d$ and by $d\ell$ in Section 6.2. The expression $d\ell$ is probably better.

  6. The authors focus mainly on the diagonalization of the Liouvillian in the four different examples. I would suggest the authors provide (e.g. in the second example) the time-evolution and/or steady state of the density matrix that correspond to the parametrization (42) of the Liouvillian, in order to provide a slightly more physical picture.

  7. While Lindblad master equations to describe open quantum system dynamics are common and relevant in many situations, it is sometimes necessary to go beyond the standard Born-Markov approximations to describe non-Markovian effects (e.g. in the solid-state). Would it be possible to generalize the flow equations to these situations (e.g. non-Markovian master equations (time non-local master equations) or more simply Redfield master equations ( time-local but non-positive master equations (see e.g. [1]) ? The authors may want to provide comments on that, as it could provide potentially more interest in the development of the method.

[1] R. Hartmann and W. T. Strunz, Accuracy assessment of perturbative master equations: Embracing nonpositivity, Phys. Rev. A 101, 012103 (2020).

---

## Round 2 · Referee Report · François Damanet (Referee 1) · 2020-10-9

Report

I thank the authors for their detailed response and the changes made. I recognise that the authors have made some effort to improve the weaknesses pointed out in the previous report.
I thus recommend the manuscript for publication in SciPost Physics.

Requested changes

-

---

## Round 2 · Referee Report · Anonymous (Referee 2) · 2020-11-26

Report

I have read the paper “Dissipative flow equations” by Lorenzo Rosso et al. where a theory generalising flow equations, first introduced for Hamiltonian systems, to open quantum dynamics is presented.
I find the paper really interesting and generally well written. The method presented is also nicely put into context and well motivated. Beyond presenting the general theory the authors discuss several examples which are of interest and of help to understand the theoretical results.

I have only one concern with the paper and this is about the notation used in section 2.
The notation does not make clear whether the transformation $\mathcal{S}$, and thus also its generator $\eta$, is a matrix of the same dimension of $\rho$ or whether it is actually a super-operator as is the Lindblad super-operator, i.e. a map acting on matrices such as $\rho$.

I try to explain myself.
In Eq.(2) $\mathcal{S}$ seems to have the same role of a Lindblad super-operator and thus one would conclude that $\mathcal{S}$ is a map acting over matrices. As such, also $\eta$ in Eq.(3) would be a similar object.
The commutators in Eq.(4) would thus be commutators between maps, i.e. the difference that one has when acting first with a map and then with the other or the other way around.
However, in Eq.(9-10) it actually seems that $\mathcal{S}$ is a matrix just like $\rho$ or $O$, even more when looking at Eq.(12), which suggests that the generator of $\mathcal{S}$, $\eta$, is a matrix like $O$. This is also the case for the last equation in pag. 5.

Also Eq.(14) seems to treat $\mathcal{S}$ as a simple matrix (by the way I think that the $\rho(t)$ inside the square bracket in the definition of $\rho_\ell (t)$ should not have a time-dependence).
However, in the line just below Eq.(14), one finds the relation $\mathcal{S}(\ell)e^{\mathcal{L}t}\mathcal{S}^{-1}(\ell)=e^{\mathcal{L}(\ell)t}$ which seems to suggest again that $\mathcal{S}$ is a super-operator as is the Lindblad, otherwise I don't see how $\mathcal{S}$ and $\mathcal{S}^{-1}$ can be brought at the exponent. However, in Eq.(16) $\eta$, which must be an object similar to $\mathcal{S}$, seems instead a matrix again.

If I look at the examples where the Lindblad is basically always represented in a doubled space, I get the feeling that $\mathcal{S}$ should indeed be a super-operator like the Lindblad. In this representation, they are both matrices but the density matrix is instead a vector.

I fear that my confusion with the notation emerges because the authors do not make clear distinction between super-operators and matrices.
One way could be to always write that the super-operator acts on a matrix. For instance, in this notation, if $\mathcal{S}$ is really a super-operator, Eq.(2) would read as

$\mathcal{L}(\ell)[X]=\mathcal{S}(\ell)\circ \mathcal{L}\circ \mathcal{S}(\ell)^{-1}[X]$

Where $\circ$ denotes composition of superoperators and $X$ indicates that the relation holds for any matrix $X$. The above equation clearly defines $\mathcal{L}(\ell)$ through its action on any $X$, as the map obtained by acting on X first with the super-operator $\mathcal{S}^{-1}(\ell)$, then with the super-operator $\mathcal{L}$ and finally with $\mathcal{S}(\ell)$. Of course, all other equations should be adjusted accordingly.
If it is true that $\mathcal{S}$ is a super-operator then I do not understand the meaning of equations 9-10 where $\mathcal{S}$ is treated as a matrix. If $\mathcal{S}$ is a superoperator I would expect Eq.9 to read as
$$
{\rm Tr}\left(\mathcal{S}^{-1}(\ell)\circ\mathcal{S}(\ell)[\rho]O\right)
$$
where the action of $\mathcal{S}^{-1}$ can then be passed to act onto $O$. I would also expect Eq. (10) to be something like
$$
\rho_0(\ell)=\mathcal{S}(\ell)[\rho]
$$

It seems like the authors have in mind that $\mathcal{S}[X]=SXS^{-1}$ (I am not sure whether this can be true?), with $S$ an appropriate matrix, and that they are a bit confusing $S$ and $\mathcal{S}$ in the presentation. Or maybe they have always in mind the notation in the doubled space but they mix up the notation in section 2 where the doubled space has not being introduced.

If $\mathcal{S}$ is instead just a matrix as $\rho$ and not a super-operator like the Lindblad I cannot make sense of Eq.(2) as it is written now.

Another way to make everything clear could be to use, from the very beginning, a representation of the Lindblad operator as a matrix and of the density matrix as a vector. In this way, no ambiguity on the role of S can emerge. This is indeed what the authors do in the examples.

I also notice that, in section 3, Eq.(17) seems to say that the generator of the flow is actually a super-operator since it is given by the commutator of the two super-operators. As such, it looks like $\mathcal{S}$ should be a super-operator.

I believe that this "issue" with the notation could generate confusion. I would advise the authors to make this clear.

After this is done, in my opinion the paper certainly deserves to be published in Scipost physics.
  • validity: -
  • significance: -
  • originality: -
  • clarity: -
  • formatting: -
  • grammar: -

Author:  Leonardo Mazza  on 2020-12-24  [id 1107]

(in reply to Report 2 on 2020-11-26)
Category:
answer to question

Both the generator eta and S are superoperators. They act on operators and return operators. As the Referee points out, there must not be confusion about this and instead our Sec. 2.2 contains a few typos which make this message unclear. We apologize for that and thank the referee for her/his remarks, which have helped us to improve the article (as well as to find a few typos). The Sec. 2.2 has been accordingly rewritten.
A possible further source of confusion is the fact that the equations discussed in Sec. 2.2 are not those employed in the example, last subsection of Sec. 5.2. Indeed, the superfermion formalism “forces” us to treat the observable as a superoperator, in a way that is very similar to the Lindbladian. This produces slightly different equations (although of very intuitive form). We wrote a remark also in this section to prevent any possible confusion.

---

## Round 2 · Referee Report · Götz Uhrig (Referee 3) · 2020-11-30

Strengths

1
Timely, interesting issue
2
Advocating a novel approach to Lindbladians, transferred from
Hamiltonians.

Weaknesses

1
Renormalizing property of generators not discussed properly

2
Interpretation of eigenvalues of Lindbladian not complete
in terms of physical significance.

3
Incomplete representation of literature

4
Some weaknesses of presentation (line widths, misleading wording, upper/lower case letters)

Report

The authors examine whether some standard infinitesimal
generators of unitary flows, which were introduced for simplifying Hamiltonians and their dynamics, can also be used for Lindbladians. Given the rising interest in dealing with open quantum systems, the issue is interesting and timely. Althoug the approach is only tested for special, solvable cases the
submitted manuscript gets a relevant message across and deserves publication.

There are some points which can still be improved to increase the
impact of the paper. They are given below:

1)
End of page 1, beginning of page 2: In the list of physics issues for which flow equations have been used, quantum magnetism, high order perturbation theory and bound states ought to be mentioned as well, e.g., by citing
Knetter et al., EPJB 13, 209 (2000); PRL 87, 167204 (2001)
Windt et al. PRL 87, 127002 (2001);
Powalski et al. SciPost 4, 1 (2018)

2)
The remark after Eq. (16) is misleading as it suggests that there is a fundamental difference between the approach for Hamiltonians and for Lindbladians. But for both of them, one can either transform all operators in the new basis or transform H or L, respectively, forward and backward. Hence there is no conceptual difference between these two applications.

3)
At several instances, the authors state that generator 3 is optimum because of its uniform convergence. But they also state that generator 2 is advantageous if truncations are necessary. This is not a side remark, but a rather crucial point:
If the method is used *without* any approximation the precise choice of generator is only a matter of convenience. Then, a fast convergence is certainly desirable for numerical implementation. If, however, the *renormalizing* property is crucial because certain terms are truncated, the matrix elements
linking states of very different eigen energies should better be transformed first. Hence, generator 1 and 2 have this conceptual asset while generator 3 does not. This should be discussed and stated
clearly.

4)
Some graphs are difficult to understand because the lines are to thin and the colors too weak due to dotted lines, e.g., Fig. 4 and Fig. 7.

5)
When eigenvalues of the Lindbladian are discussed, for instance, in Sect. 5,
their imaginary part occurs with both signs. Clearly, one sign corresponds
to exponential decay, the other to exponential growth. In an open system, however, only the decay makes sense. Hence, the question is urging why the "wrong" sign occurs and how it happens that it does not matter in physics. The authors should clarify this very important point.

6)
Related to 5), I find it most helpful it the temporal evolution
of a physical quantity, for instance an occupation, is computed
and displayed so that one sees the convergence towards the
stationary states.

7)
What is the physics of a "strongly dissipative state"?
Can you give a definition for such a state? Which signature
would an experimentalist have to look for?

8)
In the discussion of the different generators, the authors use the
term "decay" to describe the convergence towards diagonality. I find
this wording misleading since in disspative system a true physical decay
routinely occurs - but this is physics and not a property of the
mathematical basis change. Please formulate the mathematical property
a bit clearer, i.e., less ambiguously.

Minor points are:

a)
The word "modelisation" in the very first paragraph exists in French,
but not in English as far as I know.

b)
The use of lower and upper case letters in the reference got screwed - please
improve it, e.g., bec -> BEC and so on.

Requested changes

Add some additional discussions to remove weaknesses 1 and 2,
add references and improve working where appropriate, see report.

  • validity: high
  • significance: high
  • originality: high
  • clarity: good
  • formatting: excellent
  • grammar: excellent

Author:  Leonardo Mazza  on 2020-12-24  [id 1108]

(in reply to Report 3 by Götz Uhrig on 2020-11-30)

Dear Editor,

we would like to thank Prof. Uhrig for the careful reading of the manuscript and for the very constructive criticism. Here below a list of detailed answers to the comments.

1) We have modified the article accordingly.

2) We agree with the referee. After a careful rewriting of the Section following the comments of the anonymous Referee 2, we have deleted such sentence.

3) We have strengthen this discussion in the conclusions.

4) We have produced new figures 4 and 7 with a full line which are more readable.

5) The appearance of eigenvalues with positive and negative imaginary part is an artifact of the employed formalism. While Eqs. 39 and 41 are written in a form that highlights the physical significance of the eigenvalues, we need to work with the matrix in Eq. 40, that has these strange eigenvalues. Technically, the problem is due to the anticommutation of tilde{c} and tilde{c}^dagger. We have inserted a paragraph highlighting this difficulty.

6) After discussing the stationary properties of the quantum dot, we also show with some simple arguments the time-evolution of the quantum dot. The problem is simple enough to depend only on one parameter, the initial charge density of the system; we recover the known formula for the time-evolution of the dot.

7) The phenomenology of a strongly dissipative state is that of a single state that concentrates in itself the almost totality of losses. If we call N the number of modes, and gamma the typical loss rate, a strongly-dissipative state has a loss rate scaling as N*gamma, whereas all other modes have a loss rate scaling as 1/gamma, and thus are long-lived. This phenomenology is typically appearing in systems showing superradiant states (which are strongly dissipative states) and subradiant states (which are long lived). The observation of a strongly dissipative state is difficult because is decays on the most rapid time scale. On the other hand the associated weakly-dissipative states can be observed by looking at the long-time properties of the system.

8) We agree with the referee, we have modified the article accordingly using synonyms of “decay” when speaking of the flow.

a) We have introduced the word “description”.

b) Concerning the bec→BEC, the referee is right but we do not know how to act because we use bibtex, as requested by scipost, and this seems to be enforced by the scipost-bibtex style.

---

## Round 2 · Author Response

Dear Editor,

We would like to thank the referee for his/her careful reading of our manuscript and for his/her useful report on it. We append below our answers to the points raised by the referee and the relevant changes we made in the new version of the manuscript.

                                    The authors (L. Rosso, F. Iemini, M. Schirò and L. Mazza)

Answers.

  1. In the presentation of the different generators η , the authors rely a few times on a perturbative treatment involving some parameter ξ. In order to make a clearer connection to concrete physical problems, the authors may want to give a concrete physical interpretation of ξ in the case of one or two simple examples.

Answer: In general, any situation where a term of the master equation is multiplied by a small prefactor is amenable to this description. For instance, the authors Li, Petruccione and Koch of Scientific Reports 4, 4887 (2014) discuss within this framework a ring of spins with strong dissipation and a perturbatively small spin-spin coherent coupling. The same authors discuss in Phys. Rev. X 6, 021037 (2016) an open Jaynes-Cumming lattice where the spin-photon coupling is small and treated perturbatively. Appropriate referencing to this material and information has been included in the text.

  1. In the first example of section 4, I find that the explanation of the procedure to obtain the results plotted in Fig. 1-4 could be clearer.
  2. What could correspond to the non-Hermitian matrix A (a Liouvillian or a non-Hermitian Hamiltonian ?) ?
  3. What 'flow equations' presented above are concretely solved ? Some references to the equations could be useful.
  4. the 'flow step' in Section 4 is denoted by d and by dℓ in Section 6.2. The expression dℓ is probably better.

Answer: - The use of flow equations is not limited to physics and it has a more general interest, since it just proposes a method to put in diagonal form a matrix (flow equations diagonalizing Hermitian operators are known since the XIX century). Our work has obvious applications in physics, but it could just be seen as a mathematical tool for diagonalizing non-Hermitian matrices; this is what we do in the first example. One could of course interpret the matrix A as a non-Hermitian Hamiltonian, but we discourage this because it’s just a randomly generated matrix and it probably does not satisfy physical constraints.

  • The different figures 1, 2 and 3 show the solution of the flow equations using the three different generators. Explicit reference to the generators is made in the captions to Eq. 17, 22 and 29. The new version of the text details more explicitly what we did.

  • We have modified the draft accordingly.

  • The authors focus mainly on the diagonalization of the Liouvillian in the four different examples. I would suggest the authors provide (e.g. in the second example) the time-evolution and/or steady state of the density matrix that correspond to the parametrization (42) of the Liouvillian, in order to provide a slightly more physical picture.

Answer: We have added a discussion of how to derive steady-state properties for the dissipation properties in the second example.

  1. While Lindblad master equations to describe open quantum system dynamics are common and relevant in many situations, it is sometimes necessary to go beyond the standard Born-Markov approximations to describe non-Markovian effects (e.g. in the solid-state). Would it be possible to generalize the flow equations to these situations (e.g. non-Markovian master equations (time non-local master equations) or more simply Redfield master equations ( time-local but non-positive master equations (see e.g. [1]) ? The authors may want to provide comments on that, as it could provide potentially more interest in the development of the method.

Answer: We thank the referee for this interesting suggestion. At the moment we do not think that retardation effects can be efficiently taken into account because the flow equations are a technique to obtain the eigenvalues and eigenvectors of the generator of the dynamics. This information is useful when the generator of the dynamics is local in time and time-independent. On the other hand, the Redfield equation pointed out by the reviewer seems to be within the possibilities of our method because the generator of the dynamics is a linear operator that is local in time and time independent. We have briefly mentioned this possibility in the conclusions. In general, however, one can always reintroduce the explicit description of the modes of the bath and treat the problem with the standard flow equations for the Hamiltonian, as it was already done in the ‘90s (see the cited literature in the article).

---

## Round 2 · List of Changes

List of changes:

• Subsection 3.3: Sentence added: “This situation is not uncommon … and treated perturbatively”.

• Section 4: Paragraph added: “We study the dissipative … is also presented”.

• Subsection 5.2.1 “Steady state properties” added.

• Conclusions: Sentence added: “Although … Redfield master equation”.

---

## Editorial Decision

published